# City-Adaptive Testing of Autonomous Driving with Traffic Prediction and Scenario Fuzzing

## Abstract

Autonomous Driving Systems (ADS) often struggle in complex urban environments because generic testing fails to capture city-specific traffic patterns and behaviors. To address this, we propose a city-adaptive testing framework that systematically evaluates ADS robustness by integrating spatiotemporal traffic prediction and multi-agent behavioral modeling. Our approach first introduces a novel traffic prediction model, called T-DDSTGCN, which combines graph and hypergraph representations to accurately forecast segment-level traffic speed and intersection turning probabilities. It achieves the best performance on both METR-LA and PEMS-BAY datasets, demonstrating its superior ability to capture spatiotemporal dependencies in traffic prediction tasks. Based on the predicted urban traffic flow, we construct diverse simulation scenarios enriched by a behavioral modeling framework called Primary Other Participants (POP), which simulates realistic motorcycle behavior using Level-K game theory and Social Value Orientation. To enhance scenario diversity, we further apply structured perturbations across traffic density, weather, and agent interactions. Our methodology is validated across 180 real-world urban scenarios on three industrial-scale simulation platforms, yielding 662 critical collision cases after multiple rounds of testing. We have conducted an initial manual screening of the 662 simulated accident scenarios, finding that 88.1% of these accidents closely resemble real-world accident videos and reports. Furthermore, ablation studies highlight the critical role of human-like agent behavior in exposing ADS failures. Our findings suggest that incorporating traffic context and behavioral diversity into simulation testing is crucial for ensuring ADS safety and robustness in real-world deployments.

## 1 Introduction

In August 2023, after Cruise and Waymo were authorized to deploy robotaxis in San Francisco (cpuc.ca.gov, 2023), a wave of accidents, including construction intrusions and fatal collisions, revealed how vulnerable ADS remain in complex urban settings (Tan et al., 2023). Similar challenges have been reported in other cities with different road structures and traffic cultures, such as Boston's narrow intersections or Los Angeles's fast-paced multilane highways. This indicates that existing testing pipelines often lack city adaptiveness, the ability to anticipate and handle the distinct structural, dynamic, and behavioral conditions of each urban region (Karunakaran et al., 2022; Piazzoni et al., 2022).

Simulation-based testing offers a scalable and safe alternative (Huang et al., 2016; Koopman & Wagner, 2016), but existing platforms often rely on generic or mileage-based scenarios that overlook city-specific factors. Yet real-world ADS failures are often rooted in local variations—such as intersection structures, traffic flow dynamics, and the region-specific behaviors of vulnerable road users like motorcycles (Hadj-Bachir et al., 2020; 2019). To address this, we propose a holistic city-adaptive testing framework that effectively bridges the gap between traffic forecasting and autonomous driving simulation. Unlike generic simulators that rely on random procedural generation or historical replay, our approach constructs a data-driven pipeline where a predictive model acts as a generative engine. This engine establishes a city-specific traffic baseline, capturing local flow patterns and road topologies, upon which structured fuzzing and behavioral modeling are applied. By integrating spatiotemporal traffic prediction directly with simulation testing, our framework can

systematically expose safety-critical failures in complex urban environments that generic testing paradigms often overlook.

Our framework begins by modeling city-specific traffic flows using the **T**urning-**D**ual **D**ynamic **S**patial-**T**emporal **G**raph **C**onvolution **N**etwork (T-DDSTGCN), which integrates a traffic graph and a hypergraph to capture both local and higher-order spatiotemporal dependencies. The model predicts road-segment speeds and estimates intersection turning probabilities via a heuristic Speed2Turning equation that incorporates both entry speed and speed differentials between connecting segments, enabling accurate representation of urban traffic dynamics. Building on this traffic foundation, we introduce the Primary Other Participants (POP) model to simulate realistic and potentially unsafe motorcycle behaviors, formulated with Level-K game theory and Social Value Orientation to reflect local driving tendencies. Finally, we apply scenario perturbation, a fuzzing-based method that systematically varies traffic density, environmental conditions, and agent interactions, generating diverse and challenging test scenarios. We evaluate our framework in Los Angeles and San Francisco, producing 180 city-adaptive scenarios across three simulation platforms, which resulted in 662 effective collisions. Manual inspection revealed that 88.1% of these simulated accidents closely match real-world incidents, demonstrating that combining city-specific traffic prediction, behavior modeling, and scenario perturbation significantly enhances the realism and robustness of ADS testing in complex urban environments.

In this paper, we propose a city-adaptive testing framework for evaluating ADS in urban environments. Our main contributions are:

- **Holistic City-Adaptive Framework:** We establish a unified pipeline that organically integrates city-level traffic prediction, intersection turning modeling, and ADS simulation. It serves as a bridge between machine learning-based traffic forecasting and autonomous vehicle testing, enabling the construction of simulation environments that are both structurally accurate and behaviorally realistic.

- **Generative Traffic Prediction Interface:** We deploy the T-DDSTGCN not merely as a forecasting model, but as a structural interface for scenario generation. Its graph and hypergraph architecture is specifically adapted to support downstream tasks—such as the Speed2Turning inference and sub-road flow recovery—providing a deployable and extensible foundation for reconstructing city-scale traffic flows.

- **Realistic Motorcycle Behavior Modeling:** We design the Primary Other Participants (POP) framework, which simulates human-like motorcycle maneuvers based on Level-K game theory and Social Value Orientation, generating realistic disturbance agents for robust ADS testing.

- **Structured Scenario Fuzzing:** Distinct from random parameter perturbation, we introduce a structured fuzzing mechanism driven by the predicted traffic baseline. By systematically varying traffic density and environmental conditions around realistic city-specific means, this approach generates diverse, high-risk scenarios that retain the statistical characteristics of the target urban area. Experiments on 180 city-specific scenarios in Los Angeles and San Francisco, resulting in 662 effective collision cases, validate the scalability and realism of our approach.

**Data Availability.** The code of our model is available in supplementary materials; details of scenarios and maps can be found in the Appendix.

## 2 RELATED WORK

**Traffic Flow Prediction.** Traffic flow prediction aims to capture complex spatiotemporal dependencies to forecast future traffic states. Temporal modeling has evolved from RNNs (Zhao et al., 2017) to TCNs (Li et al., 2020) and Transformers (Vaswani et al., 2017), where TCNs improve efficiency and Transformers leverage long-range self-attention. Spatial modeling progressed from grid-based CNNs (Pan et al., 2018) to GCNs (Zhang et al., 2019), with STGCN (Yu et al., 2018) and DCRNN (Li et al., 2018) capturing multi-hop and diffusion effects. Recent works, such as MT-GNN (Wu et al., 2020) and Graph WaveNet (Wu et al., 2019), introduce dynamic embeddings but often neglect higher-order dependencies.

**Turning Prediction.** Turning prediction is critical for urban traffic simulation and ADS testing. Early rule-based and statistical models (Foulaadvand & Belbasi, 2011; Liu et al., 2021) struggle to adapt to dynamic traffic conditions. Data-driven methods (Ghanim & Shaaban, 2018; Mousavizadeh et al., 2021) improve accuracy using traffic flow and probe data but often incur high computational costs. We propose the Speed2Turning equation, which efficiently estimates turning probabilities from entry speed and segment speed differences, providing a lightweight and interpretable solution for real-time intersection modeling.

**Scene Reconstruction.** One approach to scene reconstruction involves the utilization of real-world data collected from urban environments (Zhang et al., 2020; Thal et al., 2023; Zhu et al., 2023b). This data may include information on road layouts, traffic patterns, and infrastructure (Carpin et al., 2007; Medrano-Berumen & Akbaş, 2020; Tettamanti et al., 2018). By leveraging techniques such as data fusion and machine learning, researchers can process and analyze this data to generate realistic simulation scenes (Zhang et al., 2022). Another key aspect of scene reconstruction is the generation of dynamic and interactive elements within simulation scenes (Ben Abdessalem et al., 2018; Kalra & Paddock, 2016). This includes modeling the behavior of various road users, such as vehicles, pedestrians, and cyclists, as well as environmental factors such as weather conditions and road obstacles (Cheng et al., 2023; Ge et al., 2023; Priisalu et al., 2022). Researchers have developed sophisticated methods to simulate the complex interactions between these elements (Zhang & Cai, 2023), ensuring that simulation scenes accurately represent real-world scenarios.

## 3 METHOD

Our testing framework follows a city-adaptive pipeline that tailors simulation scenarios to the characteristics of specific urban areas. This approach moves beyond generic simulation environments, such as SUMO or CARLA default traffic models, by explicitly modeling three core aspects of city-level traffic dynamics: City-Specific Road Topology and Flow Patterns, Spatiotemporal Traffic Prediction, and Behavioral Disturbances via Localized Agent Modeling. By integrating these components, our city-adaptive scenario generation reconstructs urban environments that are both structurally accurate and behaviorally realistic (See Figure 1). However, bridging the gap between simulation and the real-world remains challenging. In the following subsections, we will raise and solve these challenges one by one.

### 3.1 CITY-ADAPTIVE TRAFFIC PREDICTION

We introduce **T**urning-**D**ual **D**ynamic **S**patial-**T**emporal **G**raph **C**onvolution **N**etwork (T-DDSTGCN) to forecast both road-segment speeds and intersection turning probabilities (Figure 5). The model first employs DDSTGCN (Sun et al., 2022) to capture spatiotemporal dependencies across urban road networks, leveraging graph and hypergraph structures to model both local and higher-order traffic interactions. A pooling module aggregates neighboring features, while skip connections preserve gradient flow and feature diversity in deep layers. We specifically adopt the DDSTGCN backbone because its unique dual graph-hypergraph architecture is intrinsically suited to capturing the higher-order spatial dependencies of complex intersection topologies—capabilities that standard graph models often lack. To estimate intersection maneuvers, we extend this backbone into T-DDSTGCN by engineering a generative interface that couples spatiotemporal features with our Speed2Turning heuristic. This adaptation allows the model to not only forecast segment speeds but also structurally infer turning probabilities and propagate flow to unmonitored branch roads. By iteratively updating graph and hypergraph features, T-DDSTGCN acts as the engine for our city-adaptive pipeline, transforming sparse sensor data into a comprehensive, connected traffic state for simulation.

### 3.1.1 SPEED PREDICTION

**Challenge 1: How to accurately predict the traffic flow speed on road segments?** Conventional graph-based approaches model each road segment as a node and apply GCNs to capture spatial dependencies (Wu et al., 2019; 2020). However, most focus on first-order node interactions and overlook higher-order dependencies embedded in dynamic traffic edges (Sun et al., 2022), which are critical for accurate prediction. To address this, we employ T-DDSTGCN, which jointly models the traffic graph and its dual hypergraph to capture multi-level spatiotemporal dependencies. The

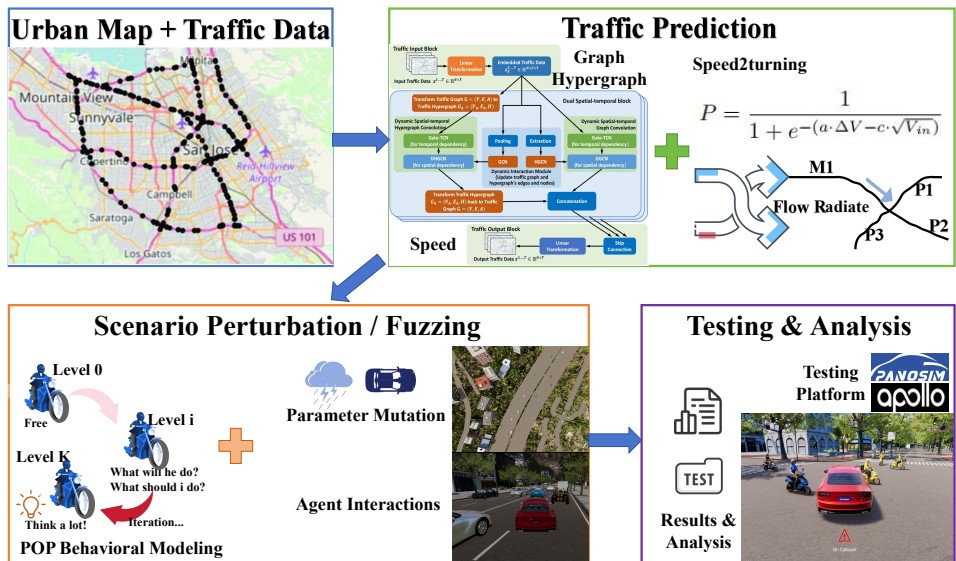

Figure 1: Overview of our City-Adaptive Testing Framework. T-DDSTGCN for traffic speed and turning predictions, followed by city-specific scene reconstruction and POP-based behavioral modeling. Scenario perturbation generates diverse, high-risk scenarios for evaluation on multiple ADS simulation platforms.

network consists of a traffic input layer, multiple Dual Spatial-Temporal (DST) Blocks, and an output layer. Within each DST-Block, traffic features are dynamically transformed between the graph and hypergraph domains, with a Dynamic Interaction Module refining edge representations. Temporal dependencies are learned via Gated Temporal Convolutions (Gate-TCN) (Chen et al., 2020), while GCNs model local node interactions and HGCNs uncover higher-order spatial relationships through hyperedges. By iteratively updating graph and hypergraph features, T-DDSTGCN achieves a deeper understanding of evolving traffic states, leading to more accurate segment speed predictions. Mathematically, these operations are:

$$GCN(X) = \sum_{n=0}^{N} (A_{forth}^n X \theta_{n,forth} + A_{back}^n X \theta_{n,back}),$$ (1)

$$HGCN(X_h) = \sum_{n=0}^{N} W_h^n X_h \theta_n.$$ (2)

where $A_{forth}^n$ and $A_{back}^n$ are the $n$-th order adjacency matrices for the forward and backward directions, and $W_h$ is the weight matrix for hyperedges in hypergraph $G_h$. The Dynamic Interaction Module (DIM) is crucial for updating the representations of edges in both the traffic graph and hypergraph. By leveraging updated node features from preceding DST-Blocks, DIM recalculates and refines edge-level features. These updated edge features are then used to dynamically adjust node features in subsequent DST-Blocks through additional GCN and HGCN operations. This iterative process creates a feedback loop, enabling the model to continuously adapt to the evolving traffic network.

**Solution 1**: We use the proposed T-DDSTGCN for traffic prediction. Simultaneously analyzing traditional traffic network graphs and their dual traffic network hypergraphs can decode and predict traffic behavior through complex analysis of the complex spatiotemporal relationships that permeate the traffic network.

### 3.1.2 SPEED2TURNING EQUATION

**Challenge 2: How can the traffic flow of branch roads be effectively monitored under the influence of sensor distribution?** Urban intersections often lack sufficient sensor coverage to

directly observe turning flows, making turning probability estimation crucial for realistic traffic scenario generation (Alexander et al., 2002; Dias et al., 2020). We introduce Speed2Turning, a lightweight heuristic that infers directional turning probabilities from predicted traffic speeds, offering a practical balance between computational efficiency and behavioral realism. Speed2Turning computes an attraction factor $P$ for each candidate turning direction using two key signals: Speed differential $\Delta V$ between entering and exiting road segments, where higher differentials indicate smoother downstream flow; Incoming speed $V_{in}$, which reflects congestion levels and influences driver turning preferences. The attraction factor is modeled with a sigmoid function:

$$P = \frac{1}{1 + e^{-(a \cdot \Delta V - c \cdot \sqrt{V_{in}})}} \quad (3) \qquad\qquad p_i = \frac{e^{P_i}}{\sum_j e^{P_j}} \quad (4)$$

where $a$ and $c$ are tunable coefficients calibrated with empirical data. To obtain normalized turning probabilities $p_i$ for all possible directions, we apply a softmax over the computed attraction factors. This formulation captures the nonlinear influence of traffic speed and congestion on driver turning behavior and can be easily adapted to different traffic conditions via coefficient calibration (More details can be seen in Appendix C).

**Solution 2**: We introduce a heuristic equation named 'Speed2Turning', which aims to effectively estimate turning probabilities using predicted traffic speeds. This formula allows the flow prediction of the branch road network to be transformed into the main road flow multiplied by the intersection's turning probability.

### 3.2 POP-ENHANCED SCENE SIMULATION AND SCENARIO FUZZING

Scene simulation builds on predicted traffic flows and real map data to reconstruct urban environments for ADS testing. Dynamic elements, such as vehicles and pedestrians, are generated from traffic flow predictions, while static elements (road geometry, lanes) come from map data. Environmental conditions, such as weather or visibility, are configured in the simulation platform to reflect local characteristics. This setup provides the foundation for scenario perturbation and fuzzing-based generation of diverse urban test cases. The scene simulation algorithm is shown in Algorithm 1.

A key challenge lies in accurately simulating motorcycles, which significantly increase the complexity of urban interactions and contribute to a large proportion of traffic accidents (DMV, 2025a; Berkeley, 2025). Unlike sensor-rich vehicles, motorcycle behavior depends heavily

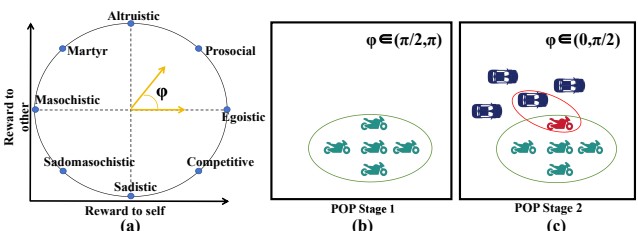

Figure 2: The POP (Primary Other Participants) Behavioral Modeling Framework. (a) SVO-Based Decision Logic: The driver's Social Value Orientation ($\varphi$) determines the weights ($\omega_1, \omega_2$) in the utility function, dictating whether the generated trajectory is aggressive (competitive) or cautious (prosocial). (b) Stage 1, Initialization: Motorcycles are initially spawned on branch roads to form a background flow without immediate conflict. (c) Stage 2, Leader-Follower Swarm Interaction: Upon entering the interaction zone, the motorcycle nearest to the ADS becomes the Stackelberg Leader (Red), optimizing its trajectory against the ADS via Level-K game theory. The remaining motorcycles act as a Follower Swarm (Green), aligning their movements with the Leader using flocking rules. This hierarchical structure simulates realistic group disturbances.

on human judgment and is often unpredictable to ADS (See Figure 7). **Challenge 3: How to accurately simulate motorcycles?** To address this, we introduce the Primary Other Participants (POP) model (See Figure 2), which combines Level-K game theory and Social Value Orientation (SVO) (Schwarting et al., 2019) to generate human-like motorcycle behaviors for interference testing. POP quantifies the degree of selfishness or cooperativeness in driver decision-making, enabling realistic simulation of aggressive or cautious motorcycle strategies. Notably, in this paper, the algorithm is designed to generate motorcycle-type traffic participants as interference groups. In our POP algorithm, integrating interactive motorcycles as interference groups involves two stages. In **stage one** (See Figure 2 part $b$), for experimental convenience, the generated motorcycle fleet initially appears only on branch roads, avoiding interaction with main roads and large-scale traffic flow. In

**stage two** (See Figure 2 part $c$), to capture the characteristic swarm-like dynamics of motorcycle groups while maintaining computational efficiency, we implement a leader-follower hierarchy. The motorcycle nearest to the ADS is designated as the interaction Leader (Stackelberg leader), optimizing its trajectory via the Level-K utility function to actively challenge the ADS. The remaining motorcycles act as a Follower Swarm, adjusting their behaviors based on the leader's movements using simplified flocking rules (e.g., alignment and separation). This approach effectively simulates complex multi-agent group disturbances without the prohibitive cost of solving a simultaneous N-player equilibrium for every agent. (more details can be seen in Appendix D)

**Solution 3**: We investigate part of ADS accident causes with a focus on modeling motorcycle behavior. Our POP model, combining game theory and social psychology, simulates motorcycles as dynamic disturbance agents, revealing long-tail risks and enhancing the realism of urban ADS testing scenarios.

---

**Algorithm 1** The whole process of scenario simulation.

---

**Input:** Urban area coordinates $\mathcal{L}$, predicted speeds $V_i$ and turning probabilities $P_i$, road network processor $\mathcal{M}$, ADS simulation platform $\mathcal{P}\text{-}\mathcal{ADS}$
**Output:** Reconstructed urban scene for ADS testing
1: Obtain OSM map for $\mathcal{L}$ and parse with $\mathcal{M}$ to generate node and link data
2: Construct and validate the road network; smooth any disconnected segments
3: Annotate road segments with predicted speeds $V_i$ and intersection turning probabilities $P_i$
4: Initialize main-road traffic flow $N$ and estimate sub-road flow $SR_j$ using::

$$SR_j = \sum_{i \in \text{incoming}(j)} P_{i \to j} \cdot N_i$$

5: Load reconstructed network into $\mathcal{P}\text{-}\mathcal{ADS}$ and verify scene validity
6: **return** Final urban scene for simulation

---

Based on the reconstructed urban scenes, which incorporate the full road network, main and branch road traffic, and initial flow predictions, we generate diverse ADS test scenarios through scenario perturbation and fuzzing. The specific fuzzing testing algorithm is outlined in Algorithm 2. Controlled variations are introduced along three dimensions: traffic density, environmental conditions, and dynamic interactions. Traffic density ranges from low-flow night conditions to peak-hour congestion, while environmental perturbations, such as rain, fog, or bright sunlight, test the robustness of ADS perception under adverse conditions. Dynamic interactions leverage the POP-model motorcycles, pedestrian crossings, and occasional vehicle malfunctions to produce realistic, high-risk scenarios that challenge ADS decision-making. Especially, we utilize DFS specifically to generate long-horizon, continuous driving routes that cover complex topological structures, such as consecutive intersections. This ensures that the ADS is tested against a coherent sequence of traffic challenges rather than isolated, disjointed road segments. Using the Los Angeles region as an example, we preprocess OpenStreetMap data to construct a continuous road network, apply T-DDSTGCN traffic predictions and the Speed2Turning model to establish baseline flows, and then apply perturbations such as density changes, POP interference, randomized vehicle positions, and weather variations. This pipeline yields a wide spectrum of city-adaptive test scenarios, enabling systematic evaluation of ADS performance and robustness under diverse and realistic urban conditions.

## 4 EVALUATION

To evaluate the effectiveness of our city-adaptive testing framework, we designed experiments that systematically incorporate city-specific traffic dynamics and behavioral disturbances. We select Los Angeles (LA) and San Francisco Bay (SFB) as our primary testbeds due to their high traffic complexity, availability of high-resolution traffic datasets, and frequent real-world autonomous driving incidents. And we investigate the following research questions:

- **RQ1**: How accurately can T-DDSTGCN predict traffic speed compared to existing models?

- **RQ2**: How effective is the 'Speed2Turning' equation in estimating turning probabilities?

- **RQ3**: To what extent do scenario fuzzing variations improve the robustness of ADS?

**Algorithm 2** Scenario Perturbation and Fuzzing Testing

**Input:** Scenario $S$ corresponding to urban areas and autonomous driving simulation platform $\mathcal{P} - \mathcal{ADS}$
**Output:** Mutated scene set $S_m$;
1: Import the original urban scene $S$ obtained from Algorithm 1 into the autonomous driving simulation platform $\mathcal{P} - \mathcal{ADS}$;
2: Import the autonomous driving algorithm that needs to be tested as the main vehicle and set the parameters of the main vehicle sensors;
3: Using the traffic flow data $N_r$ from the original map as regular flow, setting different coefficients to obtain the range of traffic flow during valley $N_v$ and peak $N_p$ periods;

$$e.g., N_v = 0.6 \cdot N_r, N_p = 1.5 \cdot N_r$$

4: Set up traffic participants of non motorized vehicle types participating in interactions in the scene, including pedestrians and motorcycles;
5: Adjust the weather environment of the scene;
6: Randomly select a point in the road network as the starting point for the main vehicle, and traverse the entire road network with DFS (Depth First Search) algorithm;
7: Repeat the above steps to obtain the mutated scene set $S_m$;

Table 1: The traffic speed prediction results of different methods on METR-LA and PEMS-BAY

| Models | METR-LA | | | | | | PEMS-BAY | | | | | |
| | 15 min | | | 60 min | | | 15 min | | | 60 min | | |
| | MAE | RMSE | MAPE | MAE | RMSE | MAPE | MAE | RMSE | MAPE | MAE | RMSE | MAPE |
|---|---|---|---|---|---|---|---|---|---|---|---|---|
| ARIMA | 3.99 | 8.21 | 9.60 | 6.90 | 13.23 | 17.40 | 1.62 | 3.30 | 3.50 | 3.38 | 6.50 | 8.30 |
| SAE | 4.65 | 8.74 | 9.93 | 6.67 | 11.34 | 16.19 | 1.83 | 3.27 | 3.57 | 3.19 | 6.37 | 7.92 |
| DD-PC | 3.36 | 7.15 | 9.13 | 4.35 | 9.82 | 14.01 | 2.00 | 4.12 | 4.63 | 2.19 | 4.56 | 5.50 |
| VAR | 4.22 | 7.89 | 10.20 | 6.52 | 10.11 | 15.80 | 1.74 | 3.16 | 3.60 | 2.93 | 5.44 | 6.50 |
| LSTM | 3.44 | 6.30 | 9.60 | 4.37 | 8.69 | 13.20 | 2.05 | 4.19 | 4.80 | 2.37 | 4.96 | 5.57 |
| STGCN | 2.89 | 5.76 | 7.63 | 4.61 | 9.37 | 12.68 | 1.37 | 2.95 | 2.86 | 2.51 | 5.72 | 5.81 |
| ASTGCN | 4.83 | 9.25 | 9.14 | 3.59 | 7.47 | 10.42 | 2.14 | 4.37 | 4.93 | 3.21 | 6.79 | 8.51 |
| STSGCN | 3.34 | 7.63 | 8.11 | 5.07 | 11.69 | 12.93 | 1.93 | 4.14 | 4.97 | 2.53 | 5.71 | 5.82 |
| GMAN | 2.81 | 5.57 | 7.42 | 3.43 | 7.34 | 10.01 | 1.34 | 2.93 | 2.83 | 2.47 | 5.67 | 5.73 |
| MTGNN | 2.68 | 5.17 | 6.92 | 3.47 | 7.21 | 9.93 | 1.29 | 2.86 | 2.79 | 2.46 | 5.54 | 5.67 |
| G-WaveNet | 2.69 | 5.15 | 6.9 | 3.53 | 7.37 | 10.02 | 1.28 | 2.89 | 2.74 | 2.45 | 5.56 | 5.64 |
| GTS | 2.67 | 5.27 | 7.21 | 3.46 | 7.31 | 9.93 | 1.28 | 2.84 | 2.76 | 2.29 | 5.34 | 5.47 |
| SAGDFN | **2.62** | 5.03 | **6.63** | **3.44** | 7.21 | **9.65** | **1.27** | 2.79 | 2.73 | 2.16 | 5.17 | 5.24 |
| T-DDSTGCN | 2.64 | **5.01** | 6.71 | **3.44** | **7.13** | 9.74 | **1.27** | **2.71** | **2.69** | **1.89** | **4.67** | **4.76** |

## 4.1 SETUP FOR EXPERIMENTS

For traffic speed prediction, we use two widely adopted datasets, METR-LA and PEMS-BAY, which record 5-minute interval traffic speeds from road sensors in Los Angeles and the San Francisco Bay Area. Traffic graphs are constructed based on segment distances, and datasets are split into 70%/10%/20% for training, validation, and testing. We evaluate our T-DDSTGCN model against diverse baselines, including statistical models (ARIMA (Shumway & Stoffer, 2025), VAR (Akkaya, 2021)), neural networks (SAE (Zhao et al., 2019), DD-PC (Liu et al., 2020), LSTM (Zhao et al., 2017)), and state-of-the-art graph-based models (STGCN (Yu et al., 2018), ASTGCN (Guo et al., 2019), STSGCN (Wang et al., 2021), GMAN (Zheng et al., 2020), MTGNN (Wu et al., 2020), G-WaveNet (Wu et al., 2019), GTS (Shang et al., 2021), SAGDFN (Jiang et al., 2024b)). For urban scene simulation and ADS testing, we select high-risk regions based on California accident reports (Berkeley, 2025) and sensor distribution (Figure 8). Five subdomains with dense road networks are extracted from Los Angeles (LA) and San Francisco Bay (SFB). These reconstructed city-adaptive scenes are integrated with three distinct simulation platforms to ensure the generalizability of our testing results: Apollo 8.0 (an open-source, industrial-grade Level 4 autonomous driving stack) (Fan et al., 2018), PanoSim (utilizing its built-in commercial pilot model, xDriver) (Panosim, 2025), and Oasis Sim. This diverse setup allows us to evaluate the generated scenarios against both open-source research baselines and closed-source commercial solutions.

## 4.2 ANSWERING RQ1: TRAFFIC SPEED PREDICTION PERFORMANCE AND CASE STUDY

Our evaluation employs multiple metrics—Mean Absolute Error (MAE), Root Mean Square Error (RMSE), and Mean Absolute Percentage Error (MAPE)—to ensure a multifaceted analysis of traffic speed prediction accuracy. The three metrics are defined as below, where $\mathbf{X}_{ij}$ denotes the ground-truth values, $\hat{\mathbf{X}}_{ij}$ are the predicted values, and $|\Omega|$ is the amount of predicted entries.

$$\text{RMSE} = \sqrt{\frac{\sum_{ij \in \Omega}(\mathbf{X}_{ij} - \hat{\mathbf{X}}_{ij})^2}{|\Omega|}} \quad (5) \qquad \text{MAE} = \frac{\sum_{ij \in \Omega}|\mathbf{X}_{ij} - \hat{\mathbf{X}}_{ij}|}{|\Omega|} \quad (6) \qquad \text{MAPE} = \sum_{ij \in \Omega}\frac{|\mathbf{X}_{ij} - \hat{\mathbf{X}}_{ij}|}{|\Omega| \cdot |\mathbf{X}_{ij}|} \quad (7)$$

Across the METR-LA and PEMS-BAY datasets, we evaluated various models for short-term (15min) and long-term (60min) prediction intervals, with the results summarized in Table 1. Traditional models, such as ARIMA and VAR, demonstrated the lowest predictive accuracy. Meanwhile, neural network-based models, including SAE, DD-PC, and LSTM, showed moderate improvements in performance but remained less competitive compared to Graph Convolutional Network (GCN)-based models. GCN-based models exhibit significant advantages in capturing spatial dependencies within traffic networks. We evaluated T-DDSTGCN on the METR-LA and PEMS-BAY datasets, comparing it with traditional baseline models. As shown in Table 1, T-DDSTGCN reaches the lowest RMSE in the short-term prediction intervals and the lowest MAE and RMSE in the long-term prediction intervals of the METR-LA dataset. Meanwhile, T-DDSTGCN achieves the best performance on all metrics in both short-term and long-term prediction intervals of the PEMS-BAY dataset. This consistency validates its reliability as a stable generation engine for our testing framework. Furthermore, distinct from general-purpose forecasting models, the hypergraph-based architecture of T-DDSTGCN provides the necessary structural adaptability for our downstream Speed2Turning inference, justifying its selection as the framework's backbone.

## 4.3 ANSWERING RQ2: SPEED2TURNING EQUATION EFFECTIVENESS AND CASE STUDY

To validate the Speed2Turning equation, we analyzed its accuracy in estimating turning probabilities by comparing its results with real-world observed turning data from a four-way signalized intersection in a metropolitan area. The dataset consisted of 250,000 recorded vehicle trajectories over a period of six months, capturing variations in turning rates across different traffic conditions (Government, 2024). The collected data includes entry speeds of vehicles approaching the intersection, exit road selections for turning movements (left, right, or straight). The dataset has been preprocessed to remove anomalies such as incomplete trajectories, extreme outliers in speed values, and inconsistencies in recorded turning movements. The evaluation highlights that while the Speed2Turning equation provides a foundational approach for estimating turning probabilities, adjustments are necessary to improve alignment with real-world behavior. More details can be seen in Appendix C.

To assess the effectiveness of our Speed2Turning equation, we analyze turning probabilities at a specific junction segment in Los Angeles during distinct traffic conditions: a non-peak period at noon (12:00 PM) and a peak period at 6:00 PM on a weekday, as depicted in Figure 3. It reveals significant differences in turning probabilities between peak and non-peak periods. During peak hours, there's a notable preference for continuing straight, likely reflecting commuters heading towards downtown or residential areas. For instance, congested northbound traffic on Highway 405 during peak hours discourages left turns, aligning with the model's consideration of entry speed on traffic behavior.

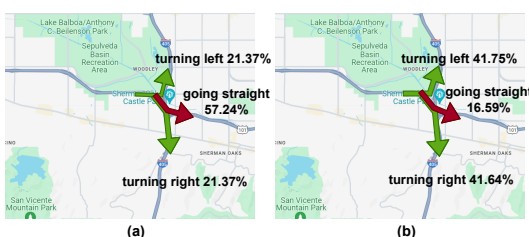

Figure 3: Probabilistic distribution of turning movements in Los Angeles. (a) observed at 18:00 on weekdays, and (b) at 12:00 on weekdays.

Table 2: Simulation and fuzzing results across varying traffic densities and environmental conditions. Each cell reports the **Average Number of Accidents** detected in **PanoSim**, followed by reproducibility indicators for **Oasis** and **Apollo**. Symbols denote: (✓) The accident scenario was successfully reproduced on the platform; (\) The platform does not support the specific scene parameters (e.g., weather settings in Apollo); (−) The accident was not reproduced.

| Urban Data | | Traffic Flow - Valley | | | Traffic Flow - Regular | | | Traffic Flow - Peak | | |
|---|---|---|---|---|---|---|---|---|---|---|
| | | Sunny | Rainy | Foggy | Sunny | Rainy | Foggy | Sunny | Rainy | Foggy |
| LA-NW | Default | 0.4✓− | 0.4−\ | 0.4−\ | 0.6✓− | 0.6✓\ | 0.8✓\ | 0.8✓✓ | 1.0✓\ | 1.0✓\ |
| | Add POP | 0.4−− | 0.4−\ | **0.6**−\ | 0.6✓− | **0.8**✓\ | **1.0**✓\ | 0.8✓✓ | 1.0✓\ | **1.6**✓\ |
| LA-CCR | Default | 0.4−− | 0.4−\ | 0.4−\ | 0.6−− | 0.6−\ | 0.6−\ | 0.8✓− | 0.8✓\ | 1.2✓\ |
| | Add POP | 0.4−− | 0.4−\ | 0.4−\ | 0.6✓− | 0.6✓\ | **0.8**✓\ | 0.8✓✓ | **1.0**✓\ | 1.2✓\ |
| LA-ECR | Default | 0.2−− | 0.2−\ | 0.2−\ | 0.4−− | 0.4−\ | 0.4−\ | 0.6✓− | 0.4−\ | 0.6✓\ |
| | Add POP | **0.4**−− | **0.4**−\ | **0.4**−\ | **0.6**✓− | **0.6**−\ | **0.6**✓\ | **1.0**✓✓ | **1.0**−\ | **1.0**✓\ |
| LA-SECR | Default | 0.2−− | 0.2−\ | 0.2−\ | 0.2−− | 0.2−\ | 0.4−\ | 0.2−− | 0.4−\ | 0.4−\ |
| | Add POP | 0.2−− | 0.2−\ | **0.4**−\ | 0.2−− | **0.4**✓\ | **0.6**✓\ | 0.2−− | 0.4−\ | **1.0**✓\ |
| LA-HW | Default | 0.6✓− | 0.6✓\ | 0.6✓\ | 1.0✓− | 1.0✓\ | 1.0✓\ | 1.4✓− | 1.6✓\ | 1.4✓\ |
| | Add POP | 0.6✓✓ | 0.6✓\ | **0.8**✓\ | 1.0✓✓ | 1.0✓\ | **1.6**✓\ | **1.6**✓✓ | 1.6✓\ | **2.6**✓\ |
| SFB-NW | Default | 1.2✓✓ | 1.2✓\ | 1.2✓\ | 1.6✓✓ | 1.6✓\ | 2.0✓\ | 2.4✓✓ | 2.6✓\ | 3.0✓\ |
| | Add POP | **1.4**✓− | 1.2✓\ | **1.4**✓\ | **2.2**✓✓ | **2.0**✓\ | **2.4**✓\ | **3.4**✓✓ | **3.2**✓\ | **3.4**✓\ |
| SFB-CA | Default | 0.4−− | 0.4−\ | 0.6−\ | 0.6✓− | 0.8✓\ | 1.0✓\ | 1.0−− | 1.0−\ | 1.6✓\ |
| | Add POP | **0.6**−− | **0.6**−\ | 0.6−\ | 0.6✓− | **1.0**−\ | 1.0✓\ | **1.6**✓✓ | **1.4**✓\ | **1.8**✓\ |
| SFB-EA | Default | 0.0−− | 0.0−\ | 0.0−\ | 0.4−− | 0.4−\ | 0.4−\ | 0.6−− | 0.4−\ | 0.6−\ |
| | Add POP | 0.0−− | 0.0−\ | **0.2**−\ | 0.4−− | 0.4−\ | **0.6**−\ | 0.6✓− | 0.4−\ | 0.4✓\ |
| SFB-SA | Default | 0.2−− | 0.2−\ | 0.2−\ | 0.4−− | 0.4−\ | 0.2−\ | 0.6✓− | 0.8✓\ | 0.6−\ |
| | Add POP | 0.2−− | **0.4**−\ | **0.4**−\ | 0.4−− | **0.6**−\ | 0.2−\ | 0.6✓✓ | 0.8✓\ | 0.6✓\ |
| SFB-NEA | Default | 0.8✓− | 0.8✓\ | 0.6✓\ | 1.0✓✓ | 1.2✓\ | 1.0✓\ | 2.2✓− | 2.0✓\ | 2.4✓\ |
| | Add POP | 0.8✓✓ | 0.8✓\ | **1.0**✓\ | **1.4**✓✓ | **1.4**✓\ | **1.8**✓\ | 2.0✓✓ | **2.2**✓\ | 2.6✓\ |

## 4.4 Answering RQ3: Effectiveness of Scenario Fuzzing and Analysis for Scene Simulation

To evaluate the impact of scenario fuzzing on ADS robustness, we conducted controlled experiments across varying traffic densities (valley, regular, peak), weather conditions (sunny, rainy, foggy), and interactive participants (default vs. POP-based). Experiments were performed using PanoSim (Panosim, 2025) on five densely monitored road network regions in Los Angeles (LA) and San Francisco Bay (SFB) to ensure realistic traffic data. For each city and configuration, the main vehicle's behavior was randomized, and each branch scenario was tested for **five rounds**. Results are summarized in Table 2, which reports the average number of ADS accidents observed in PanoSim and whether the same accident-inducing scenarios reproduced crashes on Oasis and Apollo. A check mark (✓) indicates that accidents can occur on the respective platform. A backslash (\) indicates that the platform doesn't support the corresponding scene parameters (Apollo does not support weather parameter settings).

We have summarized the experimental results as follows:

• A total of 180 test scenarios are obtained through the variation of different traffic flows, traffic participants, and other scene parameters in 10 sets of test scenarios. After 5 rounds of random setting of the main vehicle behavior, a total of 775 collision accidents are recorded. Among them, 14.58% of collision accidents are caused by setting conflicts when simulating traffic flow, with a total of 662 actual effective collision scenarios.

• In all effective collision scenarios, we observe that as the traffic density increases, the probability of collision accidents also increases. After adding the interaction participants generated by the POP algorithm, the number of accident scenarios caused by their aggressive behavior also significantly increased. After integrating interference groups generated by the POP algorithm, there is an increase in the average number of accidents in 52.2% of the test scenarios. Besides, the probability

of collision accidents varies with different weather parameters, notably increasing when visibility decreases. These results show that the deployment of ADS in central cities is indeed facing many challenges.

• In Figure 4, we present several case studies. We can import the road network structure of the corresponding city and generate urban testing scenarios under different environmental parameter configurations, like $(a)$ and $(b)$. It can also detect different types of collision accidents, where $(c)$ represents rear-end collisions caused by aggressive behavior, and $(d)$ represents collisions between autonomous vehicles and motorcycles generated by POP algorithms. Due to the randomness of testing parameters, we conduct a qualitative analysis of the causes of collision accidents. The accidents detected in the scenes we constructed encompass various factors, including incidents where autonomous vehicles are deemed responsible,

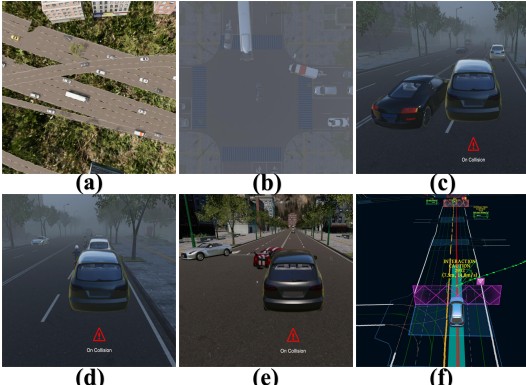

Figure 4: Simulation Result on Testing Platforms.

as well as those where other traffic participants bear responsibility. It is particularly notable that there is a substantial increase in accidents following the addition of interference groups, underscoring the significant challenges inherent in real urban environments. Figure 4 $(e)$ and $(f)$ indicate that the same testing scenario can be migrated on different testing platforms (Panosim and Apollo, respectively). These demonstrate the effectiveness and scalability of our method.

### 4.5 REALISM VALIDATION AND ABLATION STUDY

To validate the realism of our generated scenarios, we compare both driving behavior distributions and accident patterns with real-world datasets, including METR-LA, PEMS-BAY, California DMV (DMV, 2025b), NHTSA crash reports (Administration, 2025), and AV disengagement logs (Berkeley, 2025). Simulated vehicle speed distributions closely mirror real-world traffic, with mean and standard deviation deviations within ±5%, confirming that our traffic modeling accurately reproduces urban flow dynamics. Crucially, accident realism is assessed through a rigorous classification protocol: a manual review of the 662 simulated accidents confirms that **88.1%** of cases align with real-world incidents in terms of Accident Type distribution (e.g., rear-end vs. side-impact ratios) and Causal Factors (Table 6). This high statistical alignment demonstrates that our city-adaptive scenario generation captures authentic urban risk patterns rather than producing random simulation artifacts. We further conduct an ablation study to quantify each component's contribution. A fixed subset of accident scenarios is replayed under identical initial conditions to isolate the effects of traffic prediction, POP motorcycle modeling, and scenario perturbation. As shown in Table 7 and Table 8, without POP behavioral modeling, the number of discovered ADS failure rates decreases from 10.5% to 5.9%, removing scenario perturbation reduces ADS failure rates by 9.7%-25.6%, showing that fuzzing is critical for exposing diverse long-tail risks. Results confirm that each module significantly improves both scenario realism and the exposure of critical ADS failure modes. The details of the ablation experiment can also be found in the Appendix F.

## 5 CONCLUSION

We presented a city-adaptive testing framework for autonomous driving systems (ADS) that integrates spatiotemporal traffic prediction, localized behavioral modeling, and scenario perturbation to improve robustness evaluation in complex urban environments. Our approach combines three key components: T-DDSTGCN for city-specific traffic flow and turning probability prediction, POP for modeling realistic motorcycle behaviors that reflect local driving tendencies, and scenario perturbation to systematically generate diverse and challenging test scenarios. The experiments demonstrate that integrating traffic context and agent behavioral diversity is a key step toward closing the Sim2Real gap and ensuring the robustness and safety of ADS in real-world deployments.

## 6 ETHICAL DISCUSSION

This work focuses on testing and improving the robustness of autonomous driving systems through city-adaptive scenario generation and multi-agent simulation. All data used are anonymized and publicly available (e.g., OpenStreetMap, open traffic datasets). Risky behaviors are simulated purely in virtual environments to enhance system safety and are not intended for real-world replication. No human subjects or sensitive personal data are involved.

## 7 REPRODUCIBILITY STATEMENT

We provide code, data sources, and other supplementary files to ensure full reproducibility. All datasets are publicly accessible, and simulation results can be regenerated. The code of our model for predicting urban traffic flow is available at an anonymous repository `https://anonymous.4open.science/r/ASE-T-DDSTGCN-6CE4/README.md`. Details of scenarios and maps can be found in the following Appendix.

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

## A    DATA AVAILABILITY

The code of our model for predicting urban traffic flow is available at an anonymous repository `ht tps://anonymous.4open.science/r/ASE-T-DDSTGCN-6CE4/README.md`. Details of scenarios and maps can be found in the following Appendix.

## B    MORE BACKGROUND AND RELATED WORK

### B.1    RESEARCH MOTIVATION AND BACKGROUND

As an important part of ADS testing, simulation testing has irreplaceable advantages in terms of safety, controllability, and cost-effectiveness. Therefore, with the iterative development of technology, researchers are committed to deploying simulation scenarios that are closer to the real environment, from model training to finished product testing. Scenario refers to the sequence of scenarios and their interaction methods related to autonomous vehicles during the execution of dynamic driving tasks. It describes external conditions such as roads, traffic facilities, weather conditions, traffic participants, as well as information on the driving tasks and status of autonomous vehicles. Scenarios describe the complex dynamic relationship model among people, vehicles, roads, and environments in terms of space and time, which is the basis of autonomous vehicle product development and function realization. There are many methods to obtain simulation scenarios that are close to the real urban traffic environment. On the one hand, the emergence of AI Large language models has opened the door to AGI. Car companies represented by Tesla have proposed the Large World Model, which can generate future scenarios through generating models based on a large amount of real-time video data collected by autonomous vehicles, thereby achieving training and testing of models without relying on annotation information. On the other hand, researchers are also committed to collecting accident cases from real-life sources and reproducing large-scale simulation data, covering the largest possible testing space. However, these methods are too broad, and the testing method of winning by quantity is somewhat inadequate for promoting the large-scale deployment of autonomous driving technology in cities. Taking inspiration from the California case, we carefully compared the differences between closed field testing and open urban environments, and proposed a targeted regional level testing concept. Our method consists of two main parts. First, the city is regarded as a test area with a large span. Considering that the deployment of auto drive system in the middle of the city needs to meet higher safety standards, the traffic flow data of the corresponding city is used to predict the traffic flow density and other information in urban roads in different periods of time; Secondly, the predicted traffic flow information is used to restore the simulation scene of a specific city, and on this basis, scene variation is carried out to test the adaptability of the auto drive system in different urban environments.

For the first stage, we concentrate on elucidating the traffic flow characteristics within the urban road network. To achieve this, we introduce a pioneering architecture known as **T**urning-**D**ual **D**ynamic **S**patial-**T**emporal **G**raph **C**onvolution **N**etwork (T-DDSTGCN). The T-DDSTGCN framework operates through two main stages: predicting the speed of traffic flow (by deploying the DDSTGCN module) and calculating the turning probability of intersections to obtain the distribution of the entire road network. Predicting traffic speeds on road segments is a hot area of research, with numerous models proposed in recent years (Pan et al., 2018; Krupski et al., 2021; Yu et al., 2018; Guo et al., 2019). The accuracy of these predictions hinges on the model's grasp of spatial dependencies. Nevertheless, effectively capturing the dynamic dependencies inherent in traffic graphs remains challenging (Wu et al., 2020; 2019). Our adopted DDSTGCN model addresses this challenge by considering both the traffic graph and its dual, employing temporal-spatial graph convolution and temporal-spatial hypergraph neural network techniques. This comprehensive approach enables precise traffic flow prediction for road segments by accurately capturing dependencies within the traffic network (Sun et al., 2022). Moreover, for predicting turning probabilities at intersections, we introduce an innovative heuristic speed-to-turning equation. It estimates turning probabilities based on the predicted speed of vehicles entering the segment and the speed differential between entry and exit segments, enhancing the reliability of the overall traffic flow prediction model. We conduct thorough evaluations of the short-term and long-term traffic flow predictions using the DDSTGCN model, yielding improved SOTA results compared with the baseline model. Additionally, we present a case study focusing on the anticipated turning probabilities at specific intersection segments in Los Angeles. This case study serves to showcase the effectiveness of the speed-to-turning equation.

For the second stage, we leverage the previously predicted data to determine the traffic flow and speed of the corresponding urban area. However, in addition to simulating large-scale traffic flow, it's crucial to consider interference groups composed of other traffic participants, which can significantly impact the dynamics of urban traffic scenarios (Wang et al., 2022). We turn to data from fatal car and motorcycle collisions provided by the National Highway Traffic Safety Administration (of Transportation, 2023). Shockingly, over 30,843 passengers have lost their lives in such accidents in the US since 2017, accounting for 17.4% of vehicle fatalities during the same period. To simulate the real-world challenges posed by interference groups, we deliberately design a motorcycle driving behavior model based on Level-K game theory (Nagel, 1995) and SVO (Social Value Orientation) (Schwarting et al., 2019), to serve as a disturbance group in urban traffic scenarios. Subsequently, we develop a scenario fuzzing algorithm tailored for scene generation corresponding to different cities. This algorithm incorporates static information from real map data to accurately represent road layers and other environmental factors. To validate the effectiveness of our approach, we have conducted five sets of experiments for Los Angeles (LA) and San Francisco (SFB) respectively. Through these experiments, we assess the robustness of ADS across different traffic flows and urban environments. We have tested a total of 180 city scenarios and, after 5 rounds of random setting, recorded a total of 775 collision accidents, of which 662 were actually effective collision scenarios. The experimental results unequivocally demonstrate the effectiveness of our methodology in enhancing the ADS's adaptability and performance in complex urban settings.

### B.2 RELATED WORK ON TRAFFIC FLOW PREDICTION AND TURING PREDICTION

**Traffic flow prediction** aims to forecast future traffic conditions, such as vehicle speeds and traffic volumes, based on historical traffic data. The primary challenge lies in simultaneously capturing complex temporal and spatial dependencies. In addressing temporal dependencies, a variety of techniques have been widely adopted, including Recurrent Neural Networks (RNNs (Zhao et al., 2017)), Temporal Convolutional Networks (TCNs (Li et al., 2020)), and Transformers (Vaswani et al., 2017). RNNs capture long-term dependencies in time series through their recursive structure but are limited by vanishing gradient issues, making them less effective for long-sequence modeling. In contrast, TCNs efficiently handle long-range data dependencies through convolutional operations, offering superior performance in temporal modeling. Transformer-based models further improve efficiency and accuracy by leveraging self-attention mechanisms to model entire time series dependencies in parallel. In terms of spatial dimensions, earlier studies often divide the traffic network into grids and utilizing CNNs (Pan et al., 2018; Krupski et al., 2021) and their derivatives to capture spatial dependencies. However, such grid-based approaches struggle to effectively represent the complex non-Euclidean spatial structures inherent in real-world traffic networks. Recent researchers have turned to Graph Convolutional Networks (GCNs (Zhang et al., 2019)) to capture spatial dependencies among different road segments. STGCN (Yu et al., 2018) leverages multiple layers of graph convolutions to capture the spatial influence of neighboring segments over multiple hops, utilizing a first-order Chebyshev polynomial approximation for enhanced graph convolution efficiency. DCRNN (Li et al., 2018) conceptualizes traffic flow as a diffusion process on a directed graph, introducing a bidirectional random walk mechanism to effectively model spatial dependencies. However, these methods often assume a static traffic network structure, making them ill-suited for scenarios involving dynamic changes, such as traffic incidents or seasonal variations. To address the limitations of static assumptions, recent advancements, including MTGNN (Wu et al., 2020), Graph WaveNet (Wu et al., 2019), and SAGDFN (Jiang et al., 2024a) have made significant progress in this direction by utilizing continuously updated node embeddings to model spatial dependencies. Nevertheless, current dynamic graph models primarily focus on the intuitive dependencies between nodes and often overlook the higher-order dependencies that exist between edges in dynamic traffic graphs. To bridge this gap, we employ DDSTGCN, which integrates dynamic graph convolution and dynamic hypergraph convolution to achieve unified modeling of multi-level dynamic relationships between nodes and edges. This approach significantly enhances the capability to predict traffic flow in complex and evolving traffic networks.

**Turning prediction** is another critical task in urban traffic modeling, focusing on inferring vehicle turning behaviors at intersections, such as left turns, right turns, or going straight. This task typically requires integrating traffic dynamics, road geometry, and vehicle behavior features. Early methods often relied on rule-based or statistical models. (Foulaadvand & Belbasi, 2011) develops a Nagel-Schreckenberg cellular automaton model to describe vehicular traffic flow at a single intersection.

(Liu et al., 2021) addresses the uncertainty in turning ratio estimation by employing distributionally robust chance constraints. However, these methods often fail to adapt to complex and dynamic traffic scenarios. The rise of data-driven methods has introduced new perspectives for improving turning prediction. (Ghanim & Shaaban, 2018) utilizes input and output traffic flows from intersection links as inputs to an Artificial Neural Network (ANN) model, enabling flow exchange recognition at intersections without relying on prior assumptions. (Mousavizadeh et al., 2021) proposed a hybrid approach that combines sparse stationary measurements with probe vehicle data to train models for turning prediction. While these data-driven methods have shown promising results, they are often associated with high observational costs and computational delays, making them unsuitable for real-time decision-making in autonomous driving scenarios where rapid inference is crucial. To overcome these challenges, we propose an efficient and interpretable turning prediction method, the Speed2Turning equation. This approach estimates turning probabilities based on the entering speed of vehicles at intersections and the speed variation between entry and exit road segments. By modeling turning flow distributions with high efficiency, the Speed2Turning equation not only enhances the reliability of turning predictions but also provides an intuitive tool for modeling complex intersection traffic dynamics. In autonomous driving scenarios, this method enables rapid turning probability estimation at low computational costs, offering robust support for real-time traffic control in dynamic environments.

### B.3 RELATED WORK ON SCENARIO SIMULATION AND TESTING

Scenario-based testing is foundational for evaluating the performance of ADS by systematically generating diverse driving scenarios. This approach ensures comprehensive coverage of real-world conditions that an autonomous vehicle might encounter. The primary techniques in scenario-based testing include Behavior Trees, Topology-Based Scenario Classification, Bisection Method, and Data-Driven Assurance. Each technique offers a unique approach to generating test scenarios, providing comprehensive coverage and addressing specific challenges in autonomous driving.

Behavior Trees are a powerful tool for modeling the behavior of actors within a driving scenario. They allow for the precise control and description of both temporal and spatial behaviors, making them ideal for creating complex and realistic scenarios. (Han & Zhou, 2020) explores the use of behavior trees to enhance the realism and control of test scenarios. By focusing on the temporal behaviors of vehicles and pedestrians, this method improves the adaptability of scenarios in different testing environments. (Kang et al., 2022) leverages behavior trees to generate diverse and challenging scenarios for ADS, highlighting their effectiveness in varying testing contexts. Topology-Based Scenario Classification utilizes the physical layout of road networks to create diverse and representative test scenarios. This technique ensures that various road structures and conditions are covered, which is critical for comprehensive ADS testing. (Zhou et al., 2023) focuses on using road network topology to generate varied scenarios for ADS testing, ensuring comprehensive coverage of different road conditions. (Zhu et al., 2023a) discusses a method to classify junction lanes based on topology, enhancing the diversity of generated scenarios. The Bisection Method is a systematic approach to reducing the scenario space while maintaining diversity. This technique is often used in conjunction with topology-based classification to streamline the generation of diverse and challenging scenarios. (Tang et al., 2021) integrates the bisection method with topology-based classification to streamline the generation of diverse and challenging scenarios for ADS testing. Data-Driven Assurance involves creating quality criteria for parameterized scenarios to ensure they cover real traffic data instances. This approach uses search-based techniques to validate and refine test scenarios, ensuring comprehensive coverage and high quality. Scenario-based testing provides a robust framework for evaluating ADS by generating diverse and realistic driving scenarios. Techniques like behavior trees, topology-based classification, and data-driven assurance ensure comprehensive coverage and relevance, contributing significantly to the robustness and reliability of ADS testing. By systematically addressing different aspects of scenario generation, these techniques ensure that all possible driving conditions and interactions are thoroughly tested, providing a solid foundation for ADS development and validation.

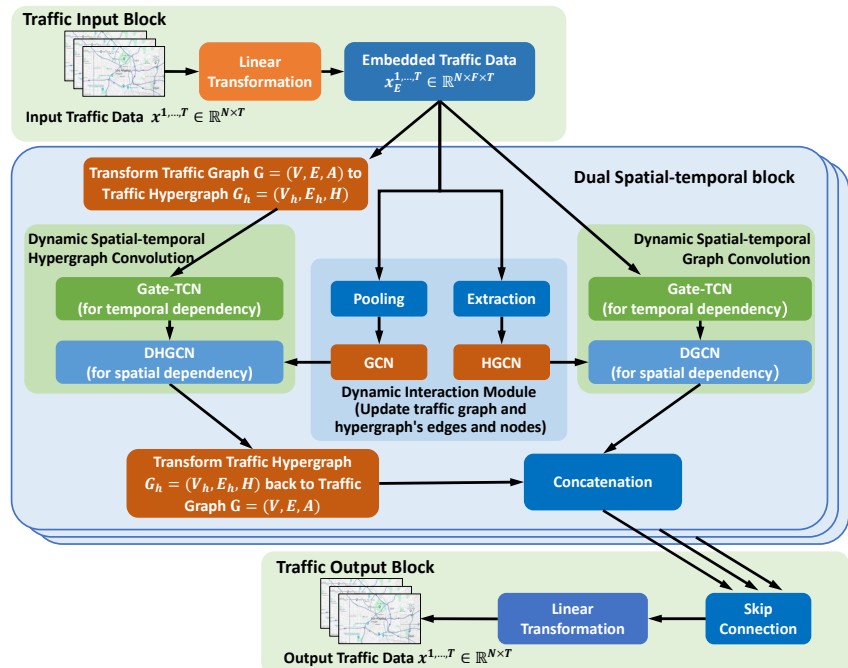

Figure 5: Framework of Dual Dynamic Spatial-Temporal Graph Convolution Network, for traffic segment speed prediction

## C   MORE DETAILS OF TRAFFIC PREDICTION

Traffic in different cities has different styles, and accurate modeling of urban traffic flow for testing plays a vital role in the deployment of ADS in cities. Open-source data offers insight into predicting traffic speed within urban road networks, yet accurately modeling the entire network's flow presents challenges. To address this, we innovatively reframe the issue by predicting vehicles' turning probabilities at intersections, effectively model branch road flow as the product of main road flow and turning probability. We introduce **T**urning-**D**ual **D**ynamic **S**patial-**T**emporal **G**raph **C**onvolution **N**etwork (T-DDSTGCN), a novel model specifically designed to forecast these turning probabilities (See Figure 5). This model initially forecasts traffic speed for each road segment leading into and out of an intersection. Subsequently, it leverages these speed predictions to calculate the likelihood of vehicles turning in various directions. For the initial step of traffic speed prediction, we incorporate DDSTGCN (Sun et al., 2022) that utilizes both the spatial graph representing physical layout of the intersection and its corresponding dual hypergraph. It can enrich the model's understanding of traffic dynamics and enable precise traffic speed forecasts. To derive turning probabilities from these speed forecasts, we propose a novel heuristic equation. This equation calculates turning probabilities based on the observed speeds of the entering road segments and the differential speeds between entering and exiting segments. This approach provides a direct method for estimating vehicle behavior at intersections which can assist in predicting traffic flow in urban road networks.

The T-DDSTGCN architecture, depicted in Fig. 5, comprises three main components: the traffic input layer, Dual Spatial-Temporal Blocks, and the traffic output layer. The Dual Spatial-Temporal Blocks facilitate the transformation of traffic data from graph to hypergraph representations, utilizing dynamic convolutions across both structures and integrating a Dynamic Interaction Module. This module continuously updates edge representations within the graph and hypergraph, allowing the DDSTGCN model to decode and predict traffic behaviors by intricately analyzing the complex spatiotemporal relationships inherent in traffic networks.

• Traffic Graph-Hypergraph Transformation.   Central to DDSTGCN is its proficient Graph-Hypergraph Transformation mechanism. This mechanism is crucial for extracting spatial dependency information from traffic flows by incorporating both the traditional traffic graph and its dual. In this dual setup, nodes in the traffic graph correspond to edges in its dual, and vice versa. This

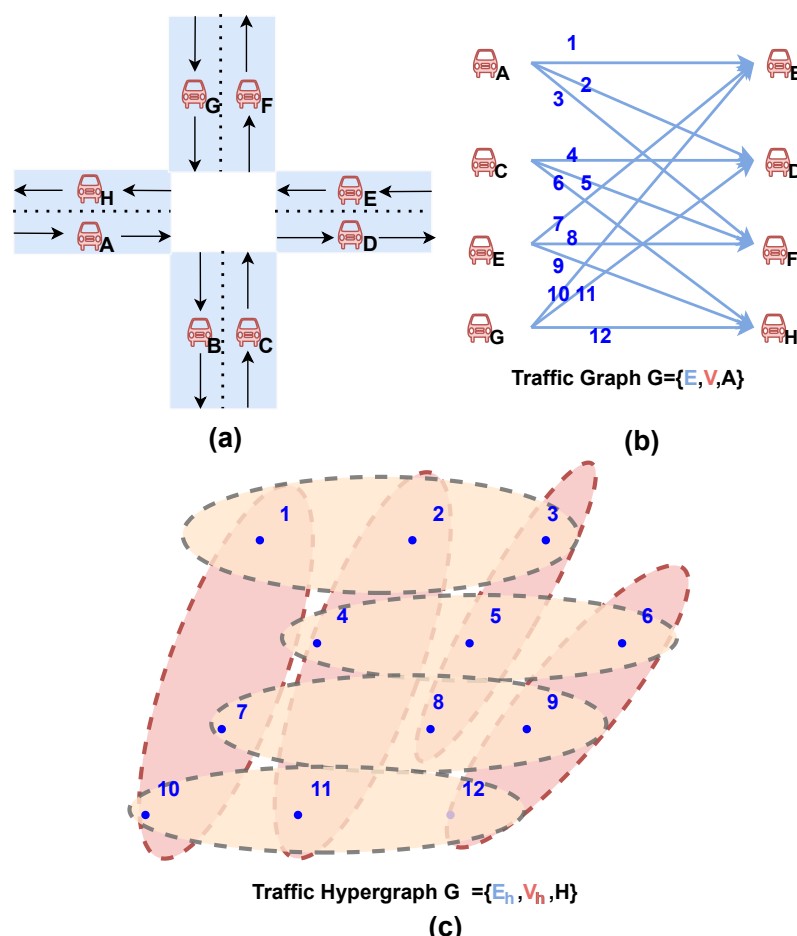

Figure 6: Example of traffic graph/hypergraph. (a) One Typical road crossing. (b) Traffic graph of this road crossing. (c) Corresponding traffic hypergraph of this road crossing. Black points in (b) and red dashed ellipses in (c) represent nodes in (a). Blue edges in (b) and blue points in (c) represent connections between nodes in (a).

mapping allows for the creation of a dual hypergraph from the original graph, where a single edge in the dual hypergraph may represent multiple nodes from the original graph, enabling it to connect more than two nodes in the original graph. This transformation is depicted in Fig.6, illustrating an example of a road intersection and highlighting both its traffic graph and corresponding hypergraph. Formally, for a traffic graph $G = (V, E, A)$ with $N$ nodes, $E$ edges, and its adjacency matrix $A$, the equivalent traffic hypergraph is denoted as $G_h = (V_h, E_h, H)$, where $|V_h| = E$, $|E_h| = N$, and $H$ is $G_h$'s incidence matrix, which is defined as:

$$H_{ij} = \begin{cases} 1 & \text{if } v_{h_i} \in e_{h_j}, \\ 0 & \text{otherwise.} \end{cases} \quad (8)$$

Given the directed nature of traffic graphs, $H$ must account for bidirectional connectivity along each edge. Consequently, $H$ is the summation of $H_{forth}$ and $H_{back}$, representing the incidence matrices for forward and backward directions, respectively. With the provided information, we can formalize the feature transformation from traffic graph to hypergraph. Given traffic graph nodes with $F$ features spanning over $T$ time periods $X \in \mathrm{R}^{N \times F \times T}$, $H_{forth}, H_{back} \in \mathrm{R}^{E \times N}$, and road distance matrix $X_{dis} \in \mathrm{R}^E$, the hyper-nodes features of $G_h$, $X_h \in \mathrm{R}^{E \times (2F+1) \times T}$, is calculated as

$$X_h = [(\theta_1 \cdot H_{forth})X | (\theta_2 \cdot H_{back})X | X_{dis}] \quad (9)$$

where $\cdot$ represents element-wise multiplication, $|$ refers to concatenation, and $\theta$ are learnable parameters. Reversely, to transform hypergraph nodes features $X'_h \in \mathrm{R}^{E \times F' \times T}$ back to graph nodes

features $X' \in \mathrm{R}^{N \times F' \times T}$, we have

$$X' = (\theta_3 \cdot H)^\top X'_h \tag{10}$$

• Dual Spatial Temporal Blocks. T-DDSTGCN incorporates an innovative sequence of Dual Spatial-Temporal Blocks (DST-Blocks) to facilitate a profound analysis of traffic data. These blocks integrate Gate-Temporal Convolutional Networks (Gate-TCN (Chen et al., 2020)), Graph Convolutional Networks (GCN (Zhang et al., 2019)), Hypergraph Convolutional Networks (HGCN (Feng et al., 2019)), and Dynamic Interaction Modules (DIM), each playing a unique role in capturing the dynamic and complex spatial-temporal patterns of traffic flow. The Gate-TCN component is specifically designed to model temporal dependencies. It captures the variations in traffic flow over time through a gating mechanism that regulates the flow of information, combined with one-dimensional convolutions across temporal dimension. The operation is formulated as:

$$TCN_\theta(X) = Conv_\theta(X) \in \mathrm{R}^{N \times F \times (T - k(T_0 - 1))} \tag{11}$$

$$G - TCN(X) = f1(TCN_{\theta 1}(X)) \cdot f2(TCN_{\theta 2}(X)) \tag{12}$$

where $k$ represents dilation factor, $T_0$ is kernal size, $f1, f2$ denote the tanh and sigmoid activation function, respectively. For capturing spatial dependencies, GCN and HGCN process the traffic graph and hypergraph, respectively. GCN focuses on direct interactions between nodes to analyze connections between road segments. In parallel, HGCN explores complex combinations of nodes, or hyperedges, to uncover hidden higher-order spatial relationships. Their operations are mathematically represented as: where $A^n_{forth}$ and $A^n_{back}$ are the $n$-th order adjacency matrices for the forward and backward directions, and $W_h$ is the weight matrix for hyperedges in hypergraph $G_h$. The Dynamic Interaction Module (DIM) is crucial for updating the representations of edges in both the traffic graph and hypergraph. By leveraging updated node features from preceding DST-Blocks, DIM processes and refreshes edge features, which, in turn, inform the dynamic updates of node features in subsequent DST-Blocks through GCN and HGCN operations.

To reconstruct urban-scale traffic flow based on sparse sensor data, we leverage the fact that most traffic sensors are deployed on major arterial roads (main roads), while minor streets or sub-roads (e.g., residential lanes) are often not instrumented. To enable flow estimation on sub-roads, we propagate traffic counts from main roads using estimated turning probabilities at intersections. Specifically, for each sub-road $SR_j$, we estimate its flow as:

$$SR_j = \sum_{i \in \mathrm{incoming}(j)} P_{i \to j} \cdot N_i$$

where: $N_i$ is the observed or estimated flow on an incoming main road segment $i$, $P_{i \to j}$ is the turning probability from road $i$ to sub-road $j$, $incoming(j)$ denotes all upstream road segments connected to $j$. This formulation allows us to approximate unobserved sub-road traffic by redistributing main-road flows based on city-specific turning behavior, which is estimated via our Speed2Turning model. The propagation step is critical in enabling city-adaptive scenario construction, as it supports traffic realism beyond the limited sensor coverage. Compared to approaches that uniformly distribute flow or use synthetic assumptions, this probabilistic mapping respects local traffic norms—e.g., a higher $P_{i \to j}$ in cities where U-turns or sharp left turns are common, or lower in regions with one-way constraints.

To validate our Speed2Turning equation, we have conducted an accuracy analysis by comparing its estimated turning probabilities with real-world observed turning data collected from a four-way signalized intersection in a metropolitan area, as shown in Table 3. The dataset consists of 250,000 recorded vehicle trajectories over a six-month period, capturing variations in turning rates under different traffic conditions. The collected data includes, entry speeds of vehicles approaching the intersections, and exit road selections for turning movements (left, right, or straight). The dataset has preprocessed to remove anomalies such as incomplete trajectories, extreme outliers in speed values, and inconsistencies in recorded turning movements. To assess the accuracy of the Speed2Turning equation, we compare its estimated turning probabilities against observed distributions using the following statistical measures.

$$KS = \sup_x |F_{pred}(x) - F_{obs}(x)| \tag{13}$$

Kolmogorov-Smirnov (KS) test measures the maximum difference between the cumulative distribution functions (CDFs) of the predicted and observed turning probabilities. A lower KS statistic

indicates a closer match between predicted and observed distributions.

$$MAE = \frac{1}{N} \sum_{i=1}^{N} |P_{pred,i} - P_{obs,i}| \tag{14}$$

Mean Absolute Error (MAE) evaluates the average absolute deviation between the predicted and observed turning probabilities.

$$RMSE = \sqrt{\frac{1}{N} \sum_{i=1}^{N} (P_{pred,i} - P_{obs,i})^2} \tag{15}$$

Root Mean Squared Error (RMSE) captures the square root of the mean of the squared deviations to emphasize larger errors.

$$r = \frac{\sum (P_{pred} - \bar{P}_{pred})(P_{obs} - \bar{P}_{obs})}{\sqrt{\sum (P_{pred} - \bar{P}_{pred})^2} \sqrt{\sum (P_{obs} - \bar{P}_{obs})^2}} \tag{16}$$

Pearson Correlation Coefficient (r) measures the strength and direction of the linear relationship between predicted and observed turning probabilities, with values close to 1 indicating a strong correlation. The Speed2Turning equation provides a reasonable approximation but tends to overestimate

Table 3: Evaluation results of Speed2Turning equation

| Metric | Left Turn | Right Turn | Straight |
|---|---|---|---|
| Observed Probability | 24.8% | 36.7% | 38.5% |
| Predicted Probability | 34.1% | 28.3% | 37.6% |
| KS Statistic | 0.28 | 0.22 | 0.14 |
| MAE | 9.3% | 8.4% | 6.1% |
| RMSE | 12.1% | 10.5% | 7.8% |
| Pearson r | 0.82 | 0.79 | 0.87 |

left turns and underestimate right turns. Observed data shows a higher prevalence of right turns, possibly influenced by dedicated turn lanes and signal timing. The model's overestimation of left turns suggests that external factors such as gaps in opposing traffic and driver caution play a significant role. The evaluation highlights that while the Speed2Turning equation provides a foundational approach for estimating turning probabilities, adjustments are necessary to improve alignment with real-world behavior. Incorporating intersection-specific parameters and real-time adaptive elements could enhance its predictive capability for autonomous driving scenario testing.

## D  POP EXTENSION MATERIALS

In current testing of autonomous driving systems, it is crucial to consider motorcycles as significant traffic participants due to their unique behavioral patterns and potential interference factors. Motorcycles, with their smaller size, higher maneuverability, and varying speeds, often present challenges for perception and decision-making algorithms. Their ability to quickly change lanes, filter through traffic, and occupy blind spots can lead to unexpected scenarios that autonomous systems must accurately detect and respond to. Additionally, as shown in Figure 7, motorcycles are involved in a significant number of traffic accidents (DMV, 2025a; Berkeley, 2025), often due to the difficulty other drivers face in predicting their movements. Incorporating motorcycles into testing scenarios ensures that these systems are robust and capable of handling real-world complexities, ultimately enhancing safety for all road users.

### D.1  BASIC DEFINITIONS

1) SVO (Social Value Orientation) (See Figure 2) quantifies the degree of selfishness exhibited by drivers, reflecting individuals' inclination toward prioritizing either their own interests or those of

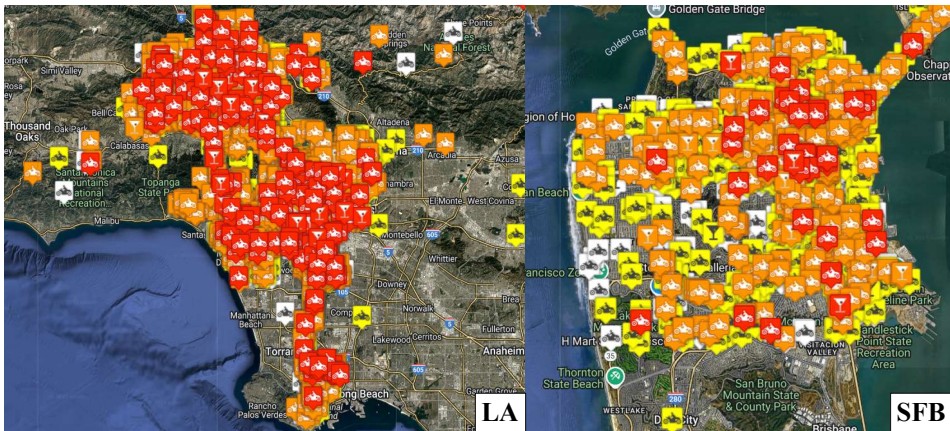

Figure 7: Motorcycle Crash Map in LA and SFB. It records 9414 formal accident reports involving motorcycles during 2020-2024, and records 3742 formal accident reports involving vehicles during 2020-2024.

others in social interactions (Schwarting et al., 2019). Unlike ADS, which relies on inference, SVO serves as an effective means to account for the impact of social suggestion and driver personality on driving behavior. 2) The Level-K game theory model first assumes that the level 0 strategy is a priori known, immature strategy that operates in a non interactive manner (Nagel, 1995). Then, a k-level driver ($k > 0$) follows a utility maximization strategy, assuming the opponent is a $(k-1)$-level driver. Starting from the 0-level strategy, the optimal strategy for k-level drivers can be recursively generated. 3) $X = [X_1^T, X_2^T, ..., X_n^T]$ refers to the state trajectory of all drivers, $U = [U_1^T, U_2^T, ..., U_n^T]$ refers to the control trajectory of all drivers, for each driver $i = 1, ..., n$, its state at time $k$ is represented as $X_i^k$, its individual control strategy is represented as $U_i^k$. In our algorithm, the utility function of driver $i$ with social attributes is defined as $g_i = \cos(\varphi_i)\omega_1 + \sin(\varphi_i)\omega_2$, where $\omega_1$ and $\omega_2$ are the 'reward to self' and 'reward to others' that defined by the driver's reward function. So the instantaneous utility function of n drivers is

$$
g_i(X, U_i, U_{\neg i}, \varphi_i) = \frac{1}{n-1} \sum_{j \in \neg i} (\cos(\varphi_i)\omega_1(X, U_i, U_j) \\
+ \sin(\varphi_i)\omega_2(X, U_j, U_i))
\tag{17}
$$

Then we discretizing the time range into T steps and accumulating the instantaneous utility encountered at each time point can obtain the cumulative utility function, which is

$$
G_i(X^0, U, \varphi) = \sum_{k=0}^{T-1} g_i(X^k, U^k, \varphi_i) + g_i^T(X^T, \varphi_i)
\tag{18}
$$

Here we define the utility of the final step as the euclidean distance from the set endpoint, so the optimization problem of the algorithm is to maximize the cumulative utility function

$$
\begin{aligned}
U_i^* &= arg\,max_{u_i} G_i(X^0, U, \varphi) \\
&= arg\,max_{u_i} (\sum_{k=0}^{T-1} g_i(X^k, U^k, \varphi_i) + g_i^T(X^T, \varphi_i)) \\
&= arg\,max_{u_i} (\sum_{k=0}^{T-1} g_i(X^k, U^k, \varphi_i) - \sqrt{(\vec{X^T} - \vec{EP})^2})
\end{aligned}
\tag{19}
$$

In our POP algorithm, integrating interactive motorcycles as interference groups involves two stages, and the kinetic data comes from official sources (DMV, 2025a). In stage one (See Figure 2 part b), for experimental convenience, the generated motorcycle fleet initially appears only on branch roads, avoiding interaction with main roads and large-scale traffic flow. These motorcycles adhere

to a fixed speed configuration, do not incorporate additional sensors, and proceed directly to their destination. In stage two (See Figure 2 part $c$), upon interaction with main vehicles, the closest driver to the main vehicle assumes the role of navigator. Their trajectory and strategy optimization are guided by maximizing a utility function, while other drivers maintain basic following strategies. A minimum safe distance between motorcycles and main vehicles is enforced, typically set at a fixed value (e.g., half a meter).

## D.2 Nash Equilibrium and Stackelberg Strategy

Nash equilibrium is a fundamental concept in game theory, named after the mathematician John Nash. It refers to a situation in a game where each player's strategy is optimal given the strategies of the other players. In other words, no player has an incentive to unilaterally change their strategy, as doing so would not lead to a better outcome for them. In game theory, Nash equilibrium is typically described using mathematical formulas. For a two-player zero-sum game (where one player's gain is exactly balanced by the other player's loss), Nash equilibrium can be defined as follows: For Player 1, their best response strategy is to maximize their expected payoff, which can be represented as: $\max_{s_1} \min_{s_2} u_1(s_1, s_2)$ For Player 2, their best response strategy is to maximize their expected payoff, which can be represented as: $\max_{s_2} \min_{s_1} u_2(s_1, s_2)$ Here, $u_1(s_1, s_2)$ and $u_2(s_1, s_2)$ represent the payoffs for Player 1 and Player 2, respectively, given the strategy combination $s_1$ and $s_2$. In Nash equilibrium, the following conditions are satisfied: $\max_{s_1} \min_{s_2} u_1(s_1, s_2) = \min_{s_2} \max_{s_1} u_1(s_1, s_2)$, $\max_{s_2} \min_{s_1} u_2(s_1, s_2) = \min_{s_1} \max_{s_2} u_2(s_1, s_2)$. In other words, at Nash equilibrium, each player's strategy maximizes their payoff given the strategies chosen by the other players.

Stackelberg strategy is a concept derived from game theory, named after German economist Heinrich Stackelberg. It is a strategic model where one player, known as the leader, makes decisions first, and then the other player, known as the follower, observes these decisions and makes their own decisions accordingly. The leader-player takes into account the anticipated response of the follower when determining their strategy, aiming to maximize their own payoff. Here's how Stackelberg strategy works and its application in decision-making: **Step 1.** Leader-Follower Dynamic: In a Stackelberg game, one player (the leader) has the advantage of moving first, while the other player (the follower) observes the leader's action before making their own decision. **Step 2.** Sequential Decision-Making: The leader makes their decision, taking into consideration the reaction of the follower. The follower, knowing the leader's decision, then selects their own strategy to maximize their payoff given the leader's action. **Step 3.** Strategic Advantage: The leader's advantage lies in their ability to anticipate and influence the follower's behavior through their initial decision. This allows the leader to strategically shape the outcome of the game in their favor. **Step 4.** Mathematical Representation: In a mathematical formulation, let $S$ denote the strategy space of the leader and $T$ denote the strategy space of the follower. The leader's payoff function is represented as $\Pi_L(S, T)$, and the follower's payoff function is represented as $\Pi_F(S, T)$. The leader aims to maximize their payoff by selecting the optimal strategy $S^*$, taking into account the follower's best response $T^*$. **Step 5.** Finding Equilibrium: The equilibrium in a Stackelberg game occurs when the leader's strategy and the follower's best response form a stable solution, where neither player has an incentive to unilaterally deviate from their chosen strategy.

## D.3 Game Theory and POP Algorithm

With the background knowledge of game theory, the foundation of our proposed POP algorithm is as follows: $X = [X_1^T, X_2^T, ..., X_n^T]$ refers to the state trajectory of all drivers, $U = [U_1^T, U_2^T, ..., U_n^T]$ refers to the control trajectory of all drivers, for each driver $i = 1, ..., n$, its state at time $k$ is represented as $X_i^k$, its individual control strategy is represented as $U_i^k$. In our algorithm, the utility function of driver $i$ with social attributes is defined as $g_i = \cos(\varphi_i)\omega_1 + \sin(\varphi_i)\omega_2$, where $\omega_1$ and $\omega_2$ are the 'reward to self' and 'reward to others' that defined by the driver's reward function. So the instantaneous utility function of n drivers is

$$
\begin{aligned}
g_i(X, U_i, U_{\neg i}, \varphi_i) = \frac{1}{n-1} \sum_{j \in \neg i} (\cos(\varphi_i)\omega_1(X, U_i, U_j) \\
+ \sin(\varphi_i)\omega_2(X, U_j, U_i))
\end{aligned}
\tag{20}
$$

Then we discretizing the time range into T steps and accumulating the instantaneous utility encountered at each time point can obtain the cumulative utility function, which is

$$G_i(X^0, U, \varphi) = \sum_{k=0}^{T-1} g_i(X^k, U^k, \varphi_i) + g_i^T(X^T, \varphi_i) \tag{21}$$

Here we define the utility of the final step as the euclidean distance from the set endpoint, so the optimization problem of the algorithm is to maximize the cumulative utility function

$$
\begin{aligned}
U_i^* &= arg\,max_{u_i} G_i(X^0, U, \varphi) \\
&= arg\,max_{u_i} (\sum_{k=0}^{T-1} g_i(X^k, U^k, \varphi_i) + g_i^T(X^T, \varphi_i)) \\
&= arg\,max_{u_i} (\sum_{k=0}^{T-1} g_i(X^k, U^k, \varphi_i) - \sqrt{(\vec{X^T} - \vec{EP})^2})
\end{aligned}
\tag{22}
$$

Considering that in actual traffic environments, there is a swarm effect among traffic participants represented by motorcycles, and most members will decide their behavior based on the leader's decision. Therefore, it is necessary to use multi-agent constraints to solve.

$$
\begin{aligned}
\mathbf{U}_1^* &= \arg\max_{\mathbf{u}_1} G_1\left(\mathbf{X}^0, \mathbf{U}_1, \mathbf{U}_2^*(\mathbf{U}_1), \varphi_1\right) \\
\text{s.t.} \quad & \mathbf{X}_1^{k+1} = \mathcal{F}_1\left(\mathbf{X}_1^k, \mathbf{U}_1^k\right) \\
& c_1\left(\mathbf{X}, \mathbf{U}_1, \mathbf{U}_2^*(\mathbf{U}_1)\right) \le 0 \\
& \mathbf{U}_2^*(\mathbf{U}_1) = \arg\max_{\mathbf{u}_2} G_2\left(\mathbf{X}^0, \mathbf{U}_1, \mathbf{U}_2, \varphi_2\right) \\
\text{s.t.} \quad & \mathbf{X}_2^{k+1} = \mathcal{F}_2\left(\mathbf{X}_2^k, \mathbf{U}_2^k\right) \\
& c_2(\mathbf{X}, \mathbf{U}) \le 0
\end{aligned}
\tag{23}
$$

This form indicates that Agent 1 can influence Agent 2's behavior by changing its own control Thus indirectly controlling the behavior of Agent 2. Considering this interaction, Agent 1 can now proactively consider how to influence Agent 2's behavior to maximize their own assistance. This involves a two-layer optimization, which involves optimizing at a higher level, including a lower level optimization problem. For each step of a high-level optimization algorithm, it is necessary to solve a low-level optimization problem. Obviously, this method cannot be extended to situations where there are more than two agents, so in this article, only the motorcycle closest to the main vehicle will be identified as the leader, and other motorcycles will be considered as a group interacting with the leader's decisions.

## E    SCENE SIMULATION EXTENSION MATERIALS

### E.1    ROAD NETWORK DATA IN THE EXPERIMENTAL URBAN AREA

Based on California's accident statistics report (Berkeley, 2025) and the distribution of road sensors, we selected the area with the highest density of sensors and the highest frequency of accidents for testing, as shown in Figure 8. We select 5 areas with dense road networks in both LA (Los Angeles) and SFB (San Francisco Bay) for experiments. The detailed data of each experimental area is shown in the Table 4.

And the road network data for 10 sets of experiments are shown in the Figure 9.

### E.2    COMPREHENSIVE SCENARIO TESTING RESULTS

Utilizing the aforementioned algorithm, we can reconstruct the initial scene of the corresponding urban area, encompassing all road network data and the initial traffic flow on main and branch roads. Building upon this foundation, parameter mutation is executed on the original scene to generate a diverse array of testing scenarios. The specific scenario fuzzing algorithm is outlined in Algorithm 2. Take the scene simulation result of Los Angeles as an example, see Figure 10. We first define

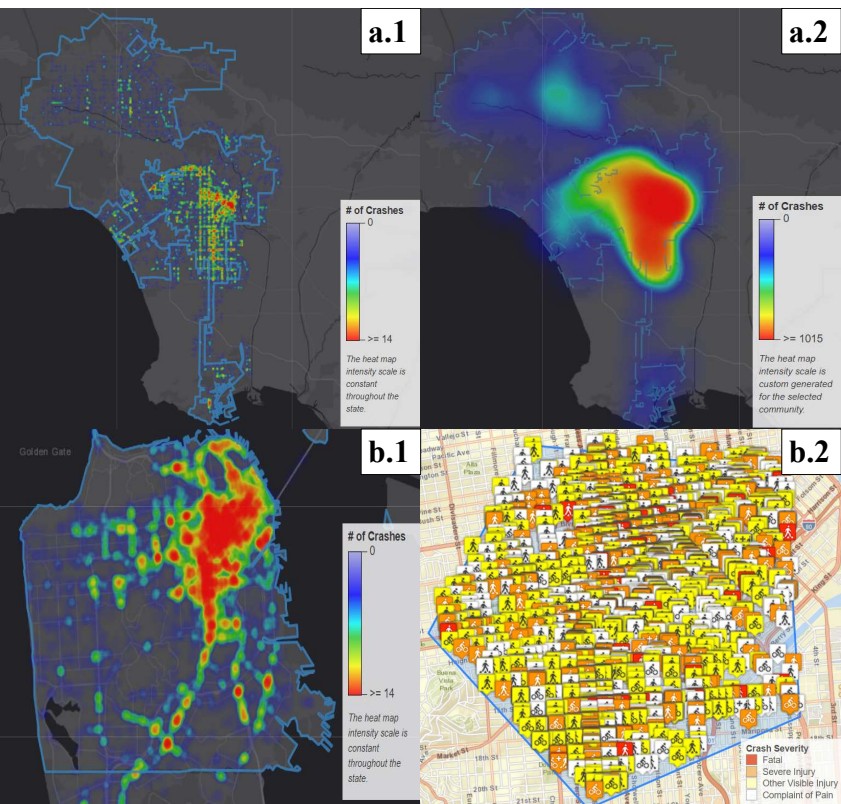

Figure 8: Number of Crashes and Community Heat Map. a.1 and a.2 show the crash heat map in LA, it records 25977 formal accident reports involving vehicles during 2018-2024, a.3 and a.4 show the crash heat map and area crash map in SFB, it records 7594 formal accident reports involving vehicles during 2018-2024.

Table 4: Simulation urban area dataset.

| Urban Data | Longitude-Left-Edge | Longitude-Right-Edge | Latitude-Top-Edge | Latitude-Bottom-Edge | Node-Num | Link-Num | Square |
|---|---|---|---|---|---|---|---|
| LA-NW | -118.4844 | -118.4636 | 34.1767 | 34.1539 | 414 | 744 | 5542 × 4066 |
| LA-CCR | -118.3859 | -118.3710 | 34.1614 | 34.1478 | 272 | 573 | 2591 × 2626 |
| LA-ECR | -118.2341 | -118.2217 | 34.1528 | 34.1401 | 301 | 498 | 4510 × 2013 |
| LA-SECR | -118.2585 | -118.2345 | 34.0662 | 34.0550 | 618 | 1125 | 3199 × 2588 |
| LA-HW | -118.3357 | -118.3133 | 34.1056 | 34.0938 | 445 | 984 | 7864 × 11927 |
| SFB-NW | -122.0730 | -122.0546 | 37.4111 | 37.3895 | 495 | 821 | 5549 × 4797 |
| SFB-CA | -122.0526 | -122.0313 | 37.3238 | 37.3098 | 388 | 698 | 3615 × 3904 |
| SFB-EA | -121.8975 | -121.8800 | 37.3340 | 37.3206 | 522 | 798 | 2723 × 3130 |
| SFB-SA | -121.9565 | -121.9383 | 37.2860 | 37.2702 | 313 | 570 | 2654 × 4447 |
| SFB-NEA | -121.8687 | -121.8528 | 37.3917 | 37.3800 | 292 | 493 | 4745 × 5107 |

• LA refers to Los Angeles, NW refers to the city's North West part, CCR refers to the city's Central Cross Road part, ECR refers to the city's Eastern Cross Road part, SECR refers to the city's Southeast Cross Road part, HW refers to the city's Hollywood part.
• SFB refers to San Francisco Bay, NW refers to the city's North West part, CA refers to the city's Central Area part, EA refers to the city's Eastern Area part, SA refers to the city's Southern Area part, NEA refers to the city's Northeast Area part.

the geographical scope of the simulation experiment, selecting the latitude and longitude range corresponding to the city under study. Then access the OpenStreetMap official website to procure the OSM map of the designated city within the specified range. Using the custom Python program, we preprocess the acquired OSM map, filtering out extraneous information to retain solely the city's road network data. This involves parsing the map to extract pertinent node and connection files, which are then inputted into the simulation platform to construct the city map. In cases where discontinuities exist within the urban road network, we employ smooth curve interpolation techniques to seamlessly connect branch roads, ensuring the network's continuity. Building upon prior predictions of traffic flow on main urban roads, we proceed to determine the speed and flow distribution at major traffic intersections. Leveraging the probability formula for speed-steering prediction, we

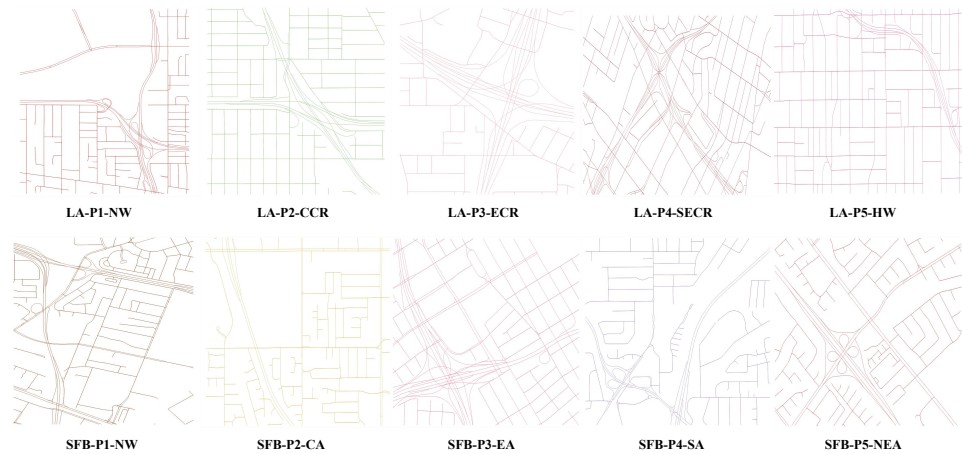

Figure 9: Road network of tested urban environments.

allocate traffic flow from main roads to branch roads, establishing a foundational urban testing scenario. Subsequently, we execute a fuzzing program within the established scenario. This involves varying the scale of traffic flow, introducing a suitable number of interference vehicles generated by the POP model, randomly selecting the starting point of the tested main vehicle, and selecting a random weather environment for the scenario. Through this process, we generate a diverse and extensive set of mutation scenarios, facilitating comprehensive testing of the ADS under various conditions.

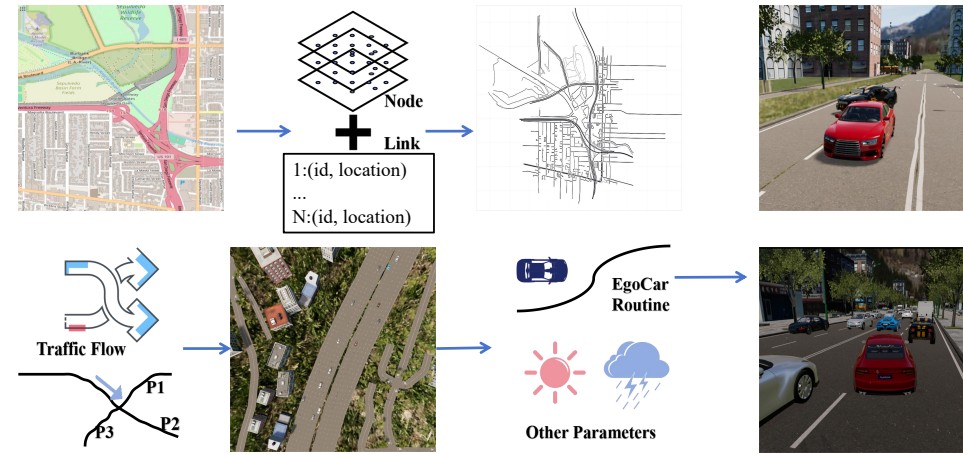

Figure 10: Scene Simulation and Scenario Fuzzing Algorithm.

In this section, we will release more experimental results, as shown in the Figure 11. Our method provides a diverse array of environmental variation parameters, as showcased in the first row of the result graph. With this approach, we can tailor urban scenes to encompass various weather conditions, including sunny, rainy, foggy, or nighttime settings. In the second row of the result graph, we observe a plethora of scene element variation parameters. Our method enables the generation of urban traffic scenes under different flow rates, with the inclusion of pedestrian and motorcycle interference groups generated by POP algorithms. The third line in the result chart presents a comprehensive array of test results for accident scenarios. Through our method, we identify a spectrum of scenarios leading to accidents in the autonomous driving system. These scenarios cover common accident modes such as lane changes and rear-end collisions, as well as conflicts with pedestrians or motorcycle interference groups generated by the POP algorithm. Collectively, these results underscore the effectiveness of our method.

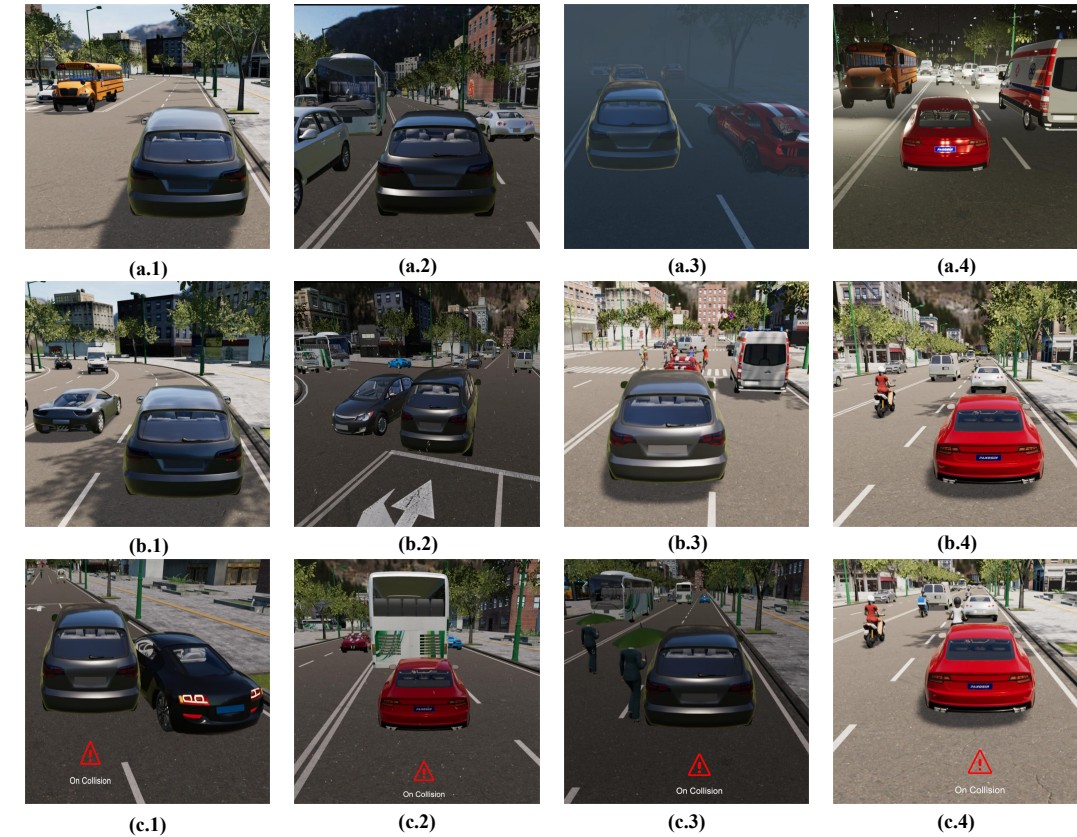

Figure 11: Comprehensive scenario testing results of various urban environments.

# F    REALISM VALIDATION AND ABLATION STUDY

## F.1    REALISM VALIDATION

To validate the realism of our generated scenarios, we compare key behavioral and accident statistics between our simulated test cases and real-world urban traffic datasets (DMV, 2025b; Administration, 2025; Berkeley, 2025). We have analyzed vehicle speed distributions in our simulation and compared them with real-world datasets collected from local transportation reports. Table 5 presents a statistical comparison. These results indicate that our simulation closely aligns with real-world

Table 5: Comparison of traffic flow metrics between real and simulated data for METR-LA and PEMS-BAY

| Metric | Real-LA | Sim-LA | Deviation(%) | Real-BAY | Sim-BAY | Deviation(%) |
|---|---|---|---|---|---|---|
| Mean Speed (m/s) | 13.2 | 12.9 | -2.3% | 14.0 | 13.6 | -2.9% |
| Speed Standard Deviation | 3.5 | 3.7 | +5.7% | 3.8 | 4.0 | +5.3% |
| Peak Hour Flow (veh/hr) | 1850 | 1805 | -2.4% | 1920 | 1875 | -2.3% |
| Off-Peak Flow (veh/hr) | 920 | 940 | +2.2% | 980 | 960 | -2.0% |

traffic flow conditions in different urban regions, with deviations within ±5% across key metrics. The small deviation in speed and flow metrics suggests that the underlying traffic models in our simulation effectively capture real-world dynamics. The slightly higher speed standard deviation in our simulation might be due to the broader range of behavioral variability introduced in synthetic

agents. Notably, the consistency in deviation percentages between METR-LA and PEMS-BAY regions suggests that our framework generalizes well to multiple urban environments.

To ensure an accurate comparison, we obtain real-world accident data from sources such as California DMV (DMV, 2025b), NHTSA crash reports (Administration, 2025), and autonomous vehicle disengagement logs from Waymo and Cruise (Berkeley, 2025). We preprocess this data by filtering relevant urban driving incidents, excluding non-traffic-related events. Then we extract accident types, severity, and cause factors for statistical aggregation. By normalizing traffic volume differences, we aim to ensure a fair comparison between real-world and simulated data. Simulated accident data is obtained from our ADS test framework, which generates diverse traffic scenarios involving varied road users and driving behaviors. The dataset includes 662 simulated accident cases, of which 88.1% (583 cases) are valid, while the remaining cases were discarded due to platform setup errors or incomplete data. The real-world dataset consists of 2,500 urban traffic accident cases, sampled to ensure diverse road conditions and accident types. Table 6 presents the accident type distribution and accident causes frequencies. The presence of overlapping accident types and

Table 6: Comparison of accident type distributions and accident causes frequencies.

| Metric | Category | Real-World Data (%) | Simulated Data (%) |
|---|---|---|---|
| **Accident Type Distribution** | Rear-End Collision | 34.5% (863/2500) | 30.0% (175/583) |
| | Side-Impact Collision | 21.7% (542/2500) | 24.0% (140/583) |
| | Intersection-Related | 38.1% (953/2500) | 28.0% (163/583) |
| | Single-Vehicle Crash | 16.2% (405/2500) | 10.0% (58/583) |
| | Pedestrian-Involved | 12.1% (302/2500) | 8.0% (47/583) |
| | Other | 8.5% (213/2500) | - |
| **Accident Causes** | Sudden Lane Change | 22.8% (570/2500) | 20.0% (117/583) |
| | Hard Braking (Rear-End) | 26.3% (658/2500) | 25.0% (146/583) |
| | Intersection Running Red Light | 25.7% (643/2500) | 22.0% (128/583) |
| | Distracted Driving | 15.9% (398/2500) | 18.0% (105/583) |
| | Speeding | 18.3% (458/2500) | 15.0% (87/583) |
| | Other | 9.4% (235/2500) | - |

● The total percentages in the Accident Type Distribution and Accident Causes category exceed 100% as multiple contributing factors can be associated with a single incident. This reflects the complexity of real-world traffic scenarios, where accidents often result from a combination of driver behaviors, environmental conditions, and roadway dynamics. The simulated data maintains a strict total of 100% due to controlled scenario configurations.

causes in real-world data (e.g., speeding combined with hard braking) results in percentages exceeding 100%, as multiple contributing factors are often present in a single incident. The distribution of accident types and causes in real-world data exhibits higher variability, reflecting the stochastic nature of urban driving conditions and external influences such as road design and traffic flow dynamics. Simulated accident distributions demonstrate strong alignment with real-world data, confirming that the scenario fuzzing methodology effectively replicates real-world risk scenarios. The close alignment of accident type distributions and accident causes demonstrates that our scenario generation effectively mimics real-world accident patterns. The minor variations are expected due to the inherent stochastic nature of human driving behavior.

## F.2 ROOT CAUSE ANALYSIS OF SIMULATED COLLISIONS

To provide deeper insights into the specific vulnerabilities of Autonomous Driving Systems (ADS) exposed by our framework, we conducted a comprehensive root cause analysis of the 662 valid collision cases. We categorize the primary failure mechanisms into three distinct classes: Behavioral Prediction Failures, Perception Degradation, and Flow Dynamics Instability.

**Behavioral Prediction Failures (Interaction-Driven)** The most prevalent failure mechanism, accounting for approximately 45% of collisions, stems from **Behavioral Prediction Failures** driven by the **POP (Primary Other Participants)** model. Current ADS often rely on conservative prediction models that assume rational, rule-abiding behavior from surrounding agents. However, our POP model, utilizing Level-K game theory and competitive SVO ($\varphi \in (0, \pi/2)$), simulates aggressive

actions such as sudden cut-ins, filtering between lanes, and forced gap acceptance. In these cases, the ADS fails to anticipate the "irrational" trajectory of the Stackelberg leader motorcycle. By the time the prediction module updates the agent's intent from "Lane Keeping" to "Cut-In," the Time-to-Collision (TTC) has often dropped below the critical braking threshold, resulting in unavoidable side-impact or oblique collisions. This underscores the fragility of rule-based prediction modules when facing socially competitive agents in complex urban settings.

**Perception Degradation (Environment-Driven)** The second major category involves **Perception Degradation**, triggered by the **Structured Scenario Fuzzing** module (approx. 30%). When scenarios are configured with adverse environmental parameters, such as rain or fog, the simulation platform introduces noise to Lidar point clouds and reduces the effective detection range of cameras. Consequently, the ADS's effective look-ahead distance is significantly compromised. In observed case studies, the system failed to detect static obstacles or slow-moving motorcycles emerging from fog until they were within emergency braking distance. This detection latency, compounded by reduced road friction coefficients modeled in the simulator under rainy conditions, frequently leads to rear-end collisions. These scenarios serve as critical regression tests for the robustness of the Perception-Control loop and validate the necessity of resilient multi-modal sensor fusion.

**Flow Dynamics Instability (Traffic-Driven)** Finally, **Flow Dynamics Instability** accounts for approximately 25% of failures, originating from the high-density traffic states predicted by **T-DDSTGCN**. In peak flow scenarios, the simulated traffic stream exhibits non-linear "stop-and-go" waves where the average spatial headway between vehicles is drastically reduced. The ADS, while proficient in steady-state car following, often struggles with these rapid oscillations in traffic speed. Specifically, when a lead vehicle performs hard braking (as modeled by the background traffic physics), the ADS's Adaptive Cruise Control (ACC) logic may exhibit delayed response times or insufficient deceleration jerk, resulting in rear-end collisions. This highlights the necessity of validating ADS performance within realistic, city-specific traffic densities rather than ideal free-flow conditions, as congestion dynamics introduce unique control challenges.

F.3    ABLATION STUDY

Given that scenario fuzzing introduces a degree of randomness, we ensure that all ablation study scenarios are conducted with the same set of traffic participants and identical initial conditions. This guarantees that performance variations are solely due to the modifications in the tested components, eliminating external variability and ensuring reliable comparisons. To assess the contributions of individual components within our framework, we have conducted an ablation study by systematically removing or modifying key elements and measuring their impact on performance. We randomly

Table 7: Impact of scenario fuzzing on ADS performance

| Scenario Variant | Accident Detection (↑) | ADS Failure (↓) | Obstacle Avoidance Success Rate (↑) |
|---|---|---|---|
| Full Scenario Fuzzing | **85.3%** | **8.2%** | **71.5%** |
| Without Environmental Variability | 80.5% | 7.4% | 72.3% |
| Without Traffic Density Variability | 82.1% | 6.8% | 75.6% |
| Without Behavioral Variability | 79.4% | 6.1% | 78.2% |

select a subset of effective accident scenarios as the testing subset for ablation experiments. The scenario fuzzing ablation (in Table 7) demonstrates that increasing the number of mutation factors increases the ADS failure rate and decreases the obstacle avoidance success rate, confirming that more complex scenarios significantly challenge ADS performance. Removing behavioral variability results in a much lower ADS failure rate and a higher avoidance success rate, suggesting that unpredictable human-like behaviors introduce critical decision-making challenges for ADS. Similarly, removing environmental variability leads to a decrease in accident detection, implying that weather and lighting variations play an important role in robust perception testing. The impact of traffic density variations further supports the necessity of testing ADS under different congestion levels, as higher density environments tend to increase failure rates due to more interactions with other vehicles and pedestrians. The ablation study on the POP motorcycle model clearly demonstrates its significant impact on ADS performance in complex urban traffic scenarios. The presence of POP motorcycles increases both the ADS failure rate and accident detection rate, as their unpredictable movements challenge the ADS's decision-making process. Without the POP model, the

Table 8: Impact of POP motorcycle model on ADS performance

| Scenario Variant | Accident Detection (↑) | ADS Failure (↓) | Obstacle Avoidance Success Rate (↑) |
|---|---|---|---|
| Default Vehicles & Pedestrians Only | 75.3% | 4.7% | 85.6% |
| With POP Model (Full) | **90.1%** | **10.5%** | **65.8%** |
| Without POP Model | 78.6% | 5.9% | 82.3% |
| Without Aggressive POP Behavior | 85.2% | 8.1% | 72.4% |
| Without Defensive POP Behavior | 87.4% | 9.3% | 69.7% |

accident detection rate drops significantly to 78.6%, indicating that ADS may fail to correctly predict and react to high-risk motorcycle behaviors. Introducing POP motorcycles leads to a dramatic increase in the ADS failure rate from 4.7% (default vehicles & pedestrians only) to 10.5%, emphasizing the critical role of unpredictable motorcycle behaviors in increasing scenario complexity. The obstacle avoidance success rate declines correspondingly from 85.6% to 65.8%, reinforcing the importance of including diverse traffic participants in ADS testing. When removing aggressive POP behaviors, the ADS failure rate decreases from 10.5% to 8.1%, and the obstacle avoidance success rate improves slightly to 72.4%, suggesting that high-speed lane-cutting and unexpected maneuvers introduce substantial difficulties for ADS models. Conversely, removing defensive behaviors (e.g., slow lane changes, hesitation at intersections) results in a failure rate of 9.3%, implying that defensive motorcycles still contribute to complex ADS decisions, especially in multi-agent interactions. These results demonstrate that ADS models perform significantly worse in environments that include motorcycles with diverse behaviors, validating the necessity of incorporating realistic motorcycle interactions in autonomous driving scenario testing. The high accident detection rate in full POP scenarios further supports the argument that ADS models require improved predictive and reactive capabilities to handle dynamic urban environments with motorcycles.

## G LLM USAGE STATEMENT

During the preparation of this manuscript, we made selective use of Large Language Models (LLMs), specifically OpenAI's ChatGPT, as a writing assistant for grammar correction and stylistic refinement. While all scientific contributions—including research ideas, model design, theoretical formulations, experiments, and result analysis—were conceived and executed entirely by the authors, the LLM was helpful in improving the clarity, fluency, and precision of the written presentation. The LLM did not contribute to any substantive intellectual content. Rather, it served as an intelligent editor—one that occasionally helped us rephrase complex sentences, refine paragraph transitions, or restructure explanations for better readability. We found this collaboration especially useful in enhancing the accessibility of our work to a broader audience. All final content was reviewed, validated, and approved by the authors.

