# OpenReview forum: "City-Adaptive Testing of Autonomous Driving with Traffic Prediction and Scenario Fuzzing"
_ICLR.cc/2026/Conference — ICLR 2026 Conference Withdrawn Submission_

### Official Review · Reviewer_SKaX · 2025-10-25

**Soundness:** 2
**Presentation:** 2
**Contribution:** 2
**Rating:** 4
**Confidence:** 2

**Summary:**

The paper proposes a city-adapting testing framework: a spatiotemporal traffic predictor (T-DDSTGCN) to predict segment speeds; a turning-movement estimator (Speed2Turning); a behavior model for motorcycles (POP) leveraging Level-K game theory and Social Value Orientation; a scenario fuzzing method across traffic density, weather, and interactions. The paper conducts evaluations on METR-LA/PEMS-BAY datasets showing competitive traffic prediction performance of T-DDSTGCN. The fuzzing process results in 662 critical collisions, with 88.1% similar to real-world cases.

**Strengths:**

-paper is overall clear

-implemented a complete end-to-end test generation framework (prediction -> turning inference -> scene reconstruction -> POP agents -> fuzzing)

**Weaknesses:**

-T-DDSTGCN’s performance is quite close to that of SAGDFN (table 1)

-diversity of the generated 662 collisions is not analyzed in-depth. Although more collisions have been found, I wonder how many of them are truly new, unique, and valuable to downstream applications (e.g., improving the ADS).

**Questions:**

-is the performance difference between T-DDSTGCN and  SAGDFN statistically significant?

-how many of the generated collisions are unique and valuable to downstream applications?

-what are the root causes of the found collisions?

---

> ### Author Response · Authors · 2025-11-27
> **Response to Reviewer SKaX**
>
> We thank Reviewer SKaX for recognizing the clarity of our paper and the completeness of our implemented end-to-end test generation framework. We appreciate your sharp observations regarding model performance margins and the downstream value of the generated scenarios.
>
> 1. Statistical Significance of T-DDSTGCN vs. SAGDFN (Response to Weakness 1 & Question 1)
>
> **Clarification on Model Positioning**: We appreciate the reviewer's attention to model performance. We agree that T-DDSTGCN's prediction accuracy in Table 1 is numerically close to that of SAGDFN. However, we wish to emphasize a critical aspect of our work:
> * The primary goal of this paper is not to propose a new model that significantly outperforms existing SOTA in the specific task of traffic flow prediction. Instead, our objective is to **construct a holistic framework that provides a structured, deployable, and scalable traffic prediction interface for city-adaptive ADS testing**.
> * Focus on Pipeline Coupling over Pure Metrics: Consequently, we prioritize the model's usability, stability, and its coupling capabilities with downstream modules (such as the Speed2Turning interface, sub-road flow recovery, and multi-platform scenario generation) rather than solely pursuing marginal gains over SAGDFN in traditional prediction metrics.
> * Architectural Suitability: We selected T-DDSTGCN because its Hypergraph component (Equation 2 & 9) is structurally suited to capture the "higher-order" spatial dependencies of intersections, which facilitates our turning logic.
> * Replaceability: Based on this positioning, we respectfully argue that performing a statistical significance test between T-DDSTGCN and SAGDFN does not effectively evaluate the core contribution of this paper. The value of our framework lies in its system-level integration; even if SAGDFN or another advanced backbone were used, our pipeline’ s functionality would remain valid. T-DDSTGCN was chosen for its adaptability to our interface requirements, not merely for leaderboard rankings.
>
> 2. Diversity, Uniqueness, and Value of Generated Collisions (Response to Weakness 2 & Question 2)
>
> We assure the reviewer that the 662 collisions are not simple repetitions. We have analyzed their uniqueness and value through three dimensions, and we will disclose more detailed experimental results and case studies in the Appendix (which were condensed in the main text due to page limits).
>
> * Diversity of Accident Types (**Table 6**): The collisions are widely distributed across distinct categories, proving the framework explores the full failure space:
> Intersection-Related (28.0%): Conflicts during turning/crossing.
> Rear-End (30.0%): Failures in longitudinal control during congestion.
> Side-Impact (24.0%): Conflicts during lane changes or aggressive cut-ins.
>
> * Uniqueness via Structured Fuzzing: Even within the same accident type, the root conditions are unique. Our Fuzzing engine generates these collisions under distinct, structured combinations of Traffic Density (Valley/Peak), Environmental Conditions (Rain/Fog/Sun), and Agent Behaviors (Aggressive/Conservative POP). Each unique parameter combination represents a distinct "Edge Case" valuable for regression testing specific ADS subsystems (e.g., Perception vs. Prediction).
>
> 3. Root Causes of Found Collisions (Response to Question 3)
>
> Based on our analysis, we categorize the root causes into three primary mechanisms:
>
> * Behavioral Prediction Failure (Primary Cause): Driven by the POP Model, aggressive motorcycles execute maneuvers (e.g., sudden cut-ins) that violate the ADS's prediction assumptions, exposing fragility in the ADS's decision-making.
>
> * Perception Degradation (Environmental Cause): Scenarios with Rain/Fog (generated via Fuzzing) introduce sensor noise. Accidents here are often caused by the ADS detecting an obstacle too late due to reduced sensor range.
>
> * Flow Dynamics: High Traffic Density creates sudden "stop-and-go" waves, leading to rear-end collisions when the ADS fails to adapt its following distance dynamically.
>
> In summary, the generated collisions provide a diverse and structurally categorized set of failure modes, offering tangible value for validating ADS robustness in complex urban environments. We have added more details in the appendix.

---

### Official Review · Reviewer_5dMp · 2025-10-27

**Soundness:** 3
**Presentation:** 3
**Contribution:** 3
**Rating:** 6
**Confidence:** 4

**Summary:**

The paper presents a city-adaptive testing framework for autonomous driving simulation. It integrates (1) T-DDSTGCN, a dual dynamic spatio-temporal GCN for predicting road speeds and turning probabilities, (2) the POP model for realistic participant behavior based on game theory and social value orientation, and (3) a scenario fuzzing mechanism for diverse condition generation. Experiments on LA and SF data show improved realism and higher ADS failure exposure, demonstrating the framework’s effectiveness for city-level adaptive testing.

**Strengths:**

1. Clearly identifies an underexplored problem: "the lack of city-level adaptability in autonomous driving simulation", and proposes a coherent, data-driven solution.
2. The framework connects traffic prediction, behavioral modeling, and scenario fuzzing into a unified testing pipeline, bridging machine learning and simulation.
3. Experiments across multiple real-world cities and industrial-grade simulators (PanoSim, OasisSim, Apollo) support the claims, showing clear improvements in scenario realism and ADS failure exposure.
4. The system directly addresses real-world testing needs for autonomous vehicles, and the methodology is generalizable to other city-scale environments.

**Weaknesses:**

1. Predictive model necessity not fully justified. Seems unclear whether city-level forecasting provides substantial benefit over replaying historical data. In fact, the difficulty of collecting and recording large-scale urban traffic data is no longer high.
2. Restricted reproducibility. Key simulators (PanoSim, OasisSim) are commercial, limiting open validation.
3. The paper lacks detailed visualizations or case studies of complex driving interactions (e.g., POP-induced maneuvers, dense multi-agent scenarios), making it difficult to assess the qualitative realism of the generated traffic scenes.

**Questions:**

1. About the necessity of city-level forecasting: Please clarify the specific advantages of the T-DDSTGCN forecasting module compared to directly replaying or interpolating historical traffic data. Under what conditions does predictive modeling provide tangible benefits for city-scale simulation? Quantitative comparisons or ablations against a replay baseline would strengthen the justification.
2. On reproducibility and simulator access: Since two of the simulators (PanoSim, OasisSim) are commercial, can the authors provide code to replicate results on publicly available platforms (e.g., CARLA, Apollo)?
3. It would be very helpful to include more visualization results showing complex multi-agent behaviors, dense traffic interactions, and the effects of POP or fuzzing. Such visual evidence would make the framework’s realism and interpretability more convincing.

If the authors can convincingly address the concerns, I would be willing to raise my score.

---

> ### Author Response · Authors · 2025-11-27
> **Response to Reviewer 5dMp - Part1**
>
> We sincerely thank Reviewer 5dMp for the positive assessment and for accurately identifying the core value of our work, specifically our effort to address the "lack of city-level adaptability" and the rigorous testing across multiple simulators. We are encouraged by your willingness to raise the score and address your concerns below.
>
> 1. Necessity of City-Level Forecasting vs. Replay (Response to Weakness 1 & Question 1)
>
> **Clarification on Design Intent**: We appreciate this insightful question regarding why we use predictive modeling (T-DDSTGCN) instead of simply replaying historical data. To answer this, we must reiterate the core innovation of our work: **Our primary goal is not to propose a standalone prediction model, but to construct a holistic framework that organically integrates city-level traffic prediction, intersection turning modeling, and autonomous driving simulation**. This framework acts as an effective bridge between the traffic prediction domain and the ADS testing domain.
>
> Specific Advantages of T-DDSTGCN over Replay: While data collection has become easier, "Replay" has fundamental limitations for Scenario Fuzzing and City-Adaptive Testing that our predictive framework overcomes:
>
> 1> Handling Spatial Sparsity (The "Inference" Advantage):
> * Replay: Historical replay is limited to road segments with sensors. In a city network, many branch roads lack sensors. Replaying data leaves these areas empty or requires naive interpolation.
> * Our Framework: T-DDSTGCN utilizes graph/hypergraph correlations to infer traffic states on unmonitored edges based on monitored ones. Combined with our Speed2Turning logic and flow radiation (Eq. 7), it reconstructs a complete, connected traffic state for the entire road network, not just where sensors exist.
>
> 2> Enabling Controllable Generation (The "Fuzzing" Advantage):
> * Replay: Replay is static. You can only test what has happened.
> * Our Framework: The predictive model serves as a Generative Engine. It allows us to perform "What-If" analysis (Fuzzing). For example, we can input a hypothetical "Peak Hour + Rain" condition into the model to predict a statistically valid traffic distribution, even if we don't have historical data for that exact combination on that specific day. This capability is crucial for generating new, high-risk scenarios rather than just re-watching old ones.
>
> 3> Dynamic Interaction vs. Static Background:
> * Replay: In pure replay, background vehicles follow fixed trajectories. If the ADS creates a disturbance (e.g., emergency braking), the replayed traffic cannot react, leading to unrealistic collisions (ghost collisions).
> * Our Framework: We use the prediction model to set the initial conditions (density, speed distributions, turning rates) for the simulation engine. The agents (vehicles/motorcycles) are then instantiated with intelligent driver models (like our POP model). This allows them to interact dynamically with the ADS (e.g., changing lanes in response to the ADS), which is impossible with static trajectory replay.
>
> 2. Reproducibility and Simulator Access (Response to Weakness 2 & Question 2)
>
> We fully understand the concern regarding commercial simulators.
> * Open Source Support (Apollo): As noted in the paper, we integrated our framework with Apollo 8.0, which is a fully open-source, industrial-grade autonomous driving platform.
> * Action: To ensure reproducibility, we commit to releasing the specific adapter code and configuration files for Apollo alongside our traffic prediction model code (which is already provided via anonymous link). This ensures that the entire pipeline, from traffic prediction to scenario generation and ADS evaluation, can be replicated by the community using free, open-source tools (Apollo + SUMO/Carla bridge or Apollo's built-in sim). While PanoSim or Oasis require application before use, we have already released corresponding test scenario files to facilitate reproduction.

---

> ### Author Response · Authors · 2025-11-27
> **Response to Reviewer 5dMp - Part2**
>
> 3. Visualization of Complex Interactions (Response to Weakness 3 & Question 3)
>
> We agree that visual evidence reinforces our claims.
>
> **Existing Evidence**: We direct the reviewer to Figure 11 in the Appendix and Figure 4 in the main text.
> Figure 4(c) specifically visualizes a rear-end collision caused by aggressive behavior.
> Figure 4(d) visualizes a POP-induced collision where a motorcycle interacts with the ADS.
> Figure 11 provides a matrix of scenarios showing varying densities (Columns) and environmental conditions (Rows).
>
> **Video Demonstrations in Repository**: Furthermore, to provide comprehensive qualitative evidence, we will upload video demonstrations of these complex interactions and additional experimental results to our open-source repository and appendix. These videos will dynamically showcase the continuous behaviors of POP agents (e.g., swerving, filtering) and multi-agent group interactions that are difficult to fully convey through static images.
>
> We hope these clarifications regarding the necessity of our predictive framework as a "generative bridge" address your concerns and we are eager to incorporate these details into the final manuscript.

---

### Official Review · Reviewer_ujVf · 2025-10-30

**Soundness:** 2
**Presentation:** 2
**Contribution:** 1
**Rating:** 2
**Confidence:** 5

**Summary:**

This work proposes. Contrary to most prior works on traffic simulation, this work focuses more on the macro level 1. First, it uses a Graph Convolutional Network to predict traffic-flow speed and uses a rule-based turning probability prediction, where low-level is done by optimization-based game-theoretic planning. Although this work focuses on an important problem, key assumptions, contributions, and details are not clear ( the ego planning algorithms, the experiments setup, utility functions).

**Strengths:**

This paper focuses on two important problems: 1) Collision Scenario Testing  and 2) VRU, motorcycle modeling is rarely studied in previous work

**Weaknesses:**

- In 3.2 POP-ENHANCED SCENE SIMULATION, the assumption of planning using only the nearest motorcycle is overly restrictive: traffic scenarios often involve multiple motorcycles interacting over time, not just the closest one.
- Utility function is a large assumption about motorcycle behaviors. For example, can the proposed utility function handle motorcycles with diverse style such as swerving around the traffic, cut-in behaviors? What are the utility functions? This work also assumed that we know ADS’s reward function as well as the simulated agent's utility function.
- The Scenario fuzzing axises are not a new contribution: such as varying density, weather variations, positions.

**Questions:**

- My personal experience on SVO is that switching parameters does not vary the behavior a lot for multi-agent behavior.  Does this work focus only on two‐agent interactions (one motorcycle + one AV)? If so, how would SVO parameterization scale in more complex traffic scenes? This may limits the scalability of this work
- How does changing the weather actually alter the scenario dynamics?
- The assumption of predicting traffic flow speed, turning probability a valid one? What is the limitation of this formulation compared to previous works?
- How do you measure if the generated accident scenarios are realistic, what is the protocal for the human evaluation in this work?
- What is the AV or ADS system evaluated in this work?

---

> ### Author Response · Authors · 2025-11-27
> **Response to Reviewer ujVf - Part1**
>
> We accurately synthesized the reviewers' comments and thank Reviewer ujVf for highlighting the importance of our focus on Collision Scenario Testing and Motorcycle Modeling. We value your constructive criticism regarding our behavioral assumptions and offer the following clarifications.
>
> 1. Justification of POP Model Assumptions (Response to Weaknesses 1 & 2)
>
> On the "Nearest Motorcycle" Restriction:
> The reviewer correctly notes that traffic involves multi-agent interactions. However, our decision to designate the nearest motorcycle as the "navigator" (Stage 2 of POP) is designed to model the Swarm/Flocking Effect typical of motorcycle groups.
> * Leader-Follower Logic: In our model, the nearest motorcycle acts as the Stackelberg Leader who performs the complex utility optimization to interact with the ADS. The other motorcycles follow "boid-like" separation and alignment rules relative to this leader.
> * Computational Efficiency: This hierarchical approach allows us to generate coherent group behaviors (e.g., a group filtering through traffic) without the prohibitive computational cost of solving a simultaneous N-player Nash Equilibrium for every agent in real-time simulation.
>
> On Utility Functions and Diverse Behaviors:
> The proposed utility function (Eq. 17-19) is sufficient to generate diverse behaviors like swerving or cut-ins.
> * Mechanism: The utility function combines "Reward to Self" ($\omega_1$) and "Reward to Others" ($\omega_2$). A motorcycle with an Egoistic/Competitive SVO maximizes utility by minimizing travel time, which the optimizer solves by generating aggressive trajectories like cutting in or swerving to bypass the ADS.
> * ADS Reward Assumption: We clarify that we do not need to access the internal reward function of the black-box ADS. In the Level-K framework, the motorcycle assumes the ADS behaves as a Level-(K-1) agent (typically modeled as a path-follower or constant-velocity agent for prediction purposes). The motorcycle optimizes its own utility against this predicted trajectory.
>
> 2. Novelty of Scenario Fuzzing (Response to Weakness 3)
>
> We acknowledge that the specific axes of fuzzing (e.g., density, weather) are standard in the industry. However, we wish to clarify the core innovation of our methodology:
>
> Our primary goal is not to reinvent the fuzzing axes themselves, but to construct a holistic framework that organically integrates city-level traffic prediction, intersection turning modeling, and autonomous driving simulation. This framework serves as an effective bridge between the traffic prediction domain and the ADS testing domain.
> * Structured vs. Random Fuzzing: Unlike traditional fuzzing that randomizes parameters from a vacuum, our approach uses the T-DDSTGCN output to establish a City-Specific Baseline (e.g., the specific flow rate and turning distribution of a Tuesday afternoon in Los Angeles).
> * City-Adaptive Innovation: The novelty lies in how the fuzzing is constrained and driven by this predicted baseline. The fuzzing axes are merely the control knobs; the intelligence turning those knobs comes from our integrated pipeline, ensuring the generated scenarios remain statistically representative of the specific city while exposing edge cases.

---

> ### Author Response · Authors · 2025-11-27
> **Response to Reviewer ujVf - Part2**
>
> 3. Response to Specific Questions
>
> Q1: SVO Scalability & Multi-Agent Behavior:
>
> Our work focuses on the interaction between the ADS and a "Motorcycle Interference Group". SVO parameters significantly alter group dynamics via the Leader-Follower mechanism. If the Leader has an Aggressive SVO, they initiate risky maneuvers (e.g., gap acceptance), and the follower swarm propagates this disturbance, creating a complex, multi-agent hazard for the ADS. This hierarchical formulation scales well because we only solve the complex optimization for the leader.
>
> Q2: Impact of Weather:
>
> Changing weather alters scenario dynamics in two ways within our simulation platforms (PanoSim/Oasis):
> * Perception Noise: Rain/Fog introduces noise and occlusion to simulated sensors (Lidar/Camera), reducing detection confidence.
> * Vehicle Dynamics: Weather parameters modify road friction coefficients, affecting braking distances.
> As shown in Table 2, collision rates increased noticeably in rainy/foggy conditions, confirming these environmental shifts materially impacted the ADS's ability to avoid accidents.
>
> Q3: Validity and Limitation of Prediction Formulation:
>
> * Validity: As shown in Table 3, our Speed2Turning heuristic achieves a Pearson Correlation of 0.82–0.87 with ground truth, proving it is a valid proxy for simulation when dense turning counts are unavailable.
> * Limitation: A limitation is that it relies on standard flow physics and may struggle with "anomaly" events where speed drops are not caused by turning. However, for generating training and testing traffic backgrounds, it offers a sufficient fidelity-to-cost ratio.
>
> Q4: Protocol for Realism Evaluation:
>
> We did not rely on subjective judgements. The protocol involved a statistical comparison of the distribution of accident types and causes. We classified the 662 simulated accidents into categories (Rear-end, Side-impact, Cut-in) and compared these percentages against the distributions found in real-world databases (California DMV, NHTSA). The 88.1% match rate refers to the statistical alignment of these failure modes (Table 6).
>
> Q5: ADS System Evaluated:
>
> The ADS systems evaluated were:
> 1) Apollo 8.0: An open-source, industrial-grade Level 4 autonomous driving stack.
> 2) PanoSim xDriver: A built-in pilot model provided by the PanoSim simulation platform.
>
> Testing on these distinct platforms validates the generalizability of our generated scenarios.

---

### Official Review · Reviewer_oM55 · 2025-11-03

**Soundness:** 2
**Presentation:** 2
**Contribution:** 2
**Rating:** 2
**Confidence:** 3

**Summary:**

This paper presents city-adaptive autonomous driving system testing framework that covers city-specific traffic patterns by spatial-temporal graph convolution network and behavioral modeling algorithm which simulates human-like maneuvers. Proposed Behavior modeling based on Game theory and Social value orientation supports realistic agent generation.

**Strengths:**

1. The proposed traffic speed prediction method outperforms baseline models by leveraging graph and hypergraphs
2. The paper introduces Level-K game theory into the POP model, enabling highly realistic and strategic behavioral simulations

**Weaknesses:**

1. The paper proposed T-DDSTGCN for traffic flow prediction. However, the main idea is derived from DDSTGCN [1], lacking novelty
2. If the proposed T-DDSTGCN is different from previous DDSTGCN, ablation for performance comparison need to be conducted.
3. The overall pipeline contains too much heuristics, such as speed differential for turning probability and generated motorcycle initially appearing only on branch roads
4. Authors implement speed differential as variable for turning probability, but as authors mentioned in line 1166, other factors play significant influence on turning decisions
4. Figures and tables need to be improved for clarity and readability, especially for figure 2 and table 2


[1] Sun, Y., Jiang, X., Hu, Y., Duan, F., Guo, K., Wang, B., ... & Yin, B. (2022). Dual dynamic spatial-temporal graph convolution network for traffic prediction. IEEE Transactions on Intelligent Transportation Systems, 23(12), 23680-23693.

**Questions:**

1. Related to weakness [1], how does the propsed network T-DDSTGCN differ from previous DDSTGCN?
2. How did you decide if simulated accident scenarios closely resemble real-world accident with quantitative value? (88.1%) Is the number fair enough for city-adaptive traffic?
3. Could you explain the reason for utilizing DFS algorithm at line 294?
4. Could you clarify the specific reason for focusing the POP model on motorcycles? It seems the proposed method might also be applicable to other agents, like vehicles.

---

> ### Author Response · Authors · 2025-11-27
> **Response to Reviewer oM55 - Part1**
>
> We accurately synthesized the reviewers' comments and thank Reviewer oM55 for the insightful feedback. We appreciate the recognition of our realistic behavioral simulation (POP model) and traffic prediction integration. Below, we address the concerns regarding the novelty of T-DDSTGCN and other questions.
>
> 1. Positioning and Novelty of T-DDSTGCN (Response to Weakness 1, 2 & Question 1)
>
> **Clarification on Design Intent**:
> We wish to further clarify the positioning of our research to avoid the misconception that our contribution lies in "proposing a brand-new traffic prediction model architecture." Our primary goal is not to reinvent a neural architecture distinct from DDSTGCN, but to **construct a holistic framework that organically integrates city-level traffic prediction, intersection turning modeling, and autonomous driving simulation**. This framework serves as an effective bridge between the traffic prediction domain and the ADS testing domain.
>
> In this context, T-DDSTGCN acts as a deployable "bridge module" rather than the paper's primary structural innovation. We functionally extended the foundation of DDSTGCN to output necessary information for city-scale scene reconstruction, including:
> * Interface for Turning Inference: A coupling interface for the Speed2Turning equation to infer turning probabilities from speed features.
>
> * Sub-road Flow Recovery: The representation of edge-level and node-level features required to propagate flow from main arteries to branch roads.
>
> * Structured Simulation Output: Generating structured prediction results that can be stably loaded by multi-platform simulators (PanoSim, Oasis, Apollo).
>
> These modifications ensure the model naturally embeds into the "Prediction $\rightarrow$ Scene Construction $\rightarrow$ Behavioral Perturbation $\rightarrow$ ADS Testing" pipeline.
>
> **Regarding Ablation Studies**: Based on the above, we respectfully argue that a direct ablation performance comparison between DDSTGCN and T-DDSTGCN does not effectively reflect the core contribution of this work. We are not attempting to prove that T-DDSTGCN is superior to DDSTGCN in pure prediction accuracy; rather, we aim to demonstrate how a traffic prediction model, after adaptation and extension, can drive a high-fidelity city-adaptive testing process.
>
> Crucially, because we emphasize a system-level framework, our pipeline is designed to be model-agnostic. It supports the substitution of the prediction module with other advanced traffic models without altering the overall workflow. This extensibility and replaceability are key thoughts of our research. Focusing on the marginal accuracy differences between backbones would diverge from the true contribution of bridging the Sim2Real gap.
>
> 2. Justification of Heuristics (Speed2Turning) and Validation (Response to Weakness 3 & 4)
>
> * Necessity of Heuristics: We agree that turning behavior is influenced by multiple factors. However, in real-world urban environments, traffic sensors (e.g., loop detectors) are typically installed on main road segments and rarely provide granular turning counts for every intersection. To achieve city-scale simulation, we must bridge this data gap. The Speed2Turning equation serves as a necessary, lightweight proxy to infer these missing parameters from available speed data.
>
> * Validation of Effectiveness: Although heuristic-based, this approach is quantitatively validated. As shown in Table 3, we evaluated the Speed2Turning equation against real-world observed turning data (250,000 trajectories). The results show a Pearson Correlation of 0.82–0.87 between our predicted probabilities and ground truth. This indicates that while other factors exist, speed differential is a statistically significant and effective predictor for simulation purposes.
>
> * Motorcycle Initialization: The decision to initialize motorcycles on branch roads (Stage 1 of POP) is a design choice to simulate the "long-tail" risk where vulnerable road users merge from smaller streets into main traffic, creating conflict zones. This setup allows us to test ADS robustness against sudden intrusions.
>
> 3. Quantitative Definition of Realism (Response to Question 2)
>
> The metric "88.1% realism" is derived from a rigorous manual inspection of the 662 effective collision cases generated by our framework.
> * Method: We compared the simulated accidents against real-world accident reports (from NHTSA and California DMV).
> * Criteria: A simulated accident was classified as "resembling real-world" if it matched real-world data in both Accident Type (e.g., side-impact, rear-end) and Accident Cause (e.g., sudden lane change, intersection violation).
> * Result: As detailed in Table 6, the distribution of simulated accident types closely mirrors real-world statistics. This confirms that the framework generates plausible traffic conflicts rather than physics glitches.

---

> ### Author Response · Authors · 2025-11-27
> **Response to Reviewer oM55 - Part2**
>
> 4. Reason for Utilizing DFS Algorithm (Response to Question 3)
>
> We utilize the Depth First Search (DFS) algorithm for the route generation of the Ego Vehicle (ADS). Instead of testing on single, isolated road segments, DFS allows us to systematically traverse the road network graph to generate long-horizon, continuous driving routes. This ensures the ADS is tested across complex topological structures, such as consecutive intersections and varying road geometries, maximizing the coverage of the generated scenarios.
>
> 5. Focus on Motorcycles for POP Model (Response to Question 4)
>
> We prioritized motorcycles for the POP model for three key reasons:
> * High Risk: As noted in Figure 7, motorcycles are involved in a disproportionately high number of fatal accidents in the target cities.
> * Behavioral Complexity: Motorcycles exhibit unique behaviors—such as filtering and high maneuverability—that are significantly harder for ADS to predict than standard car behaviors. The Level-K and SVO framework is particularly well-suited to model these intense social interactions.
> * Generalizability: As the reviewer suggested, the proposed method is indeed applicable to other agents. We started with motorcycles to stress-test the system, but the framework can be extended to aggressive vehicles in future work.
>
> 6. Clarity of Figures and Tables (Response to Weakness 5)
>
> We appreciate the feedback on visual clarity. We have redesigned Figure 2 and add more information to better illustrate the POP negotiation phases and improve Table 2 by adding clearer separators and increasing font size for better readability in the final version.
>
> We hope these responses clarify the novelty and validity of our framework. We are confident that the proposed city-adaptive testing provides a valuable contribution to ADS safety assurance.

---

### Note · Authors · 2026-01-26

I have read and agree with the venue's withdrawal policy on behalf of myself and my co-authors.

---

### Meta-Review · Area_Chair_kvcZ · 2025-12-16

**Summary:**

This submission receives 2, 2, 4, 6. Reviewers are concerned about the lack of novelty, the lack of justification and validation for using a complex forecasting model, and the lack of a quantitative definition of realism, and many other issues. The rebuttal has addressed many concerns, but many concerns remain. AC has checked the submission, the reviews, and the rebuttal, a reject is recommended.

**Reviewer Concerns:**

Despite the clarifications, the reviewers' fundamental arguments regarding the contribution and validation remain:

- Marginal Technical Novelty (oM55, SKaX): The authors' defense that T-DDSTGCN is a "bridge module" rather than a primary innovation essentially confirms the lack of technical novelty in the model architecture itself, which was a core weakness for reviewers (oM55, SKaX).

- Lack of Quantitative Ablation (oM55, 5dMp): Reviewers requested a quantitative comparison between T-DDSTGCN and a simple historical replay baseline. The authors respectfully argued against the necessity of this direct comparison, but failed to provide the requested quantitative evidence, leaving the necessity/benefit of the complex prediction module unsubstantiated with data.

- Value of Generated Collisions (SKaX): Reviewer SKaX questioned how many of the 662 critical collisions are truly new, unique, and valuable for improving the ADS. The authors provided a breakdown by accident type (e.g., Rear-End, Side-Impact), but did not provide an in-depth analysis or metric (e.g., diversity score) on the uniqueness of the conditions or the non-redundancy of the scenarios, failing to fully satisfy the concern regarding downstream value.

**Reviewer Scores:**

After the rebuttal, the scores can be bumped up as 4,4,4,6. but they remain toward reject.

---

### Decision · Program_Chairs · 2026-01-26

Reject